# High-resolution characterization of centriole distal appendage morphology and dynamics by correlative STORM and electron microscopy

Mathew Bowler[1,2], Dong Kong [1], Shufeng Sun[3], Rashmi Nanjundappa [4], Lauren Evans[5], Veronica Farmer[1,8], Andrew Holland [5], Moe R. Mahjoub[4,6], Haixin Sui[3,7] & Jadranka Loncarek[1]

Centrioles are vital cellular structures that form centrosomes and cilia. The formation and function of cilia depends on a set of centriole's distal appendages. In this study, we use correlative super resolution and electron microscopy to precisely determine where distal appendage proteins localize in relation to the centriole microtubules and appendage electron densities. Here we characterize a novel distal appendage protein ANKRD26 and detail, in high resolution, the initial steps of distal appendage assembly. We further show that distal appendages undergo a dramatic ultra-structural reorganization before mitosis, during which they temporarily lose outer components, while inner components maintain a nine-fold organization. Finally, using electron tomography we reveal that mammalian distal appendages associate with two centriole microtubule triplets via an elaborate filamentous base and that they appear as almost radial finger-like protrusions. Our findings challenge the traditional portrayal of mammalian distal appendage as a pinwheel-like structure that is maintained throughout mitosis.

[1] Laboratory of Protein Dynamics and Signaling, NIH/NCI/CCR, Frederick, Maryland 21702, USA. [2] Optical Microscopy and Analysis Laboratory, NIH/NCI/CCR, Frederick, Maryland 21702, USA. [3] Wadsworth Center, New York State Department of Health, Albany, NY 12201, USA. [4] Department of Medicine (Nephrology Division), Washington University, St Louis 63110 MO, USA. [5] Department of Molecular Biology & Genetics, Johns Hopkins University School of Medicine, Baltimore 21205 MD, USA. [6] Department of Cell Biology and Physiology, Washington University, St Louis 12201 MO, USA. [7] Department of Biomedical Sciences, School of Public Health, University of Albany, Albany, NY 12201, USA. [8]Present address: Department of Cell and Developmental Biology, Vanderbilt University School of Medicine, Nashville 37235 TN, USA. These authors contributed equally: Mathew Bowler, Dong Kong. Correspondence and requests for materials should be addressed to J.L. (email: jadranka.loncarek@nih.gov)

Centrioles are microtubule (MT)-based cylindrical structures. Human centrioles are ~500 nm long and exhibit proximal-distal polarity[1,2]. On their proximal ends, a centriole's wall is ~230 nm wide and built of nine MT triplets, which consist of one full MT and two partial MTs (Fig. 1). At the beginning of the cell cycle, vertebrate cycling cells have two centrioles. One of them is older and has undergone at least two cell cycles. The younger one was initiated in the previous cell cycle. The proximal end of both centrioles organizes a highly structured supramolecular matrix called the pericentriolar material (PCM)[3–5], which is the site of MT nucleation and centriole duplication. The distal end of centrioles is the assembly site of two types of electron-dense projections called distal and subdistal appendages (DAs and SDAs, respectively). Only the older centriole, which has a fully assembled distal end harbors appendages, while the distal end of the younger centriole is incomplete. Thus, the younger centriole lacks all functions associated with these structures. DAs are essential for ciliogenesis[6–8,6-12] because they mediate the attachment of ciliary vesicles to mother centrioles and their subsequent fusion with the cytoplasmic membrane. Subdistal appendages anchor MTs and position centrioles and cilia[13–15]. This generational gap between centrioles ensures that only one centriole forms a primary cilium[16].

Centriole appendages play a vital role in many cellular processes such as development, motility, signaling, and the maintenance of cellular architecture, making them indispensable for

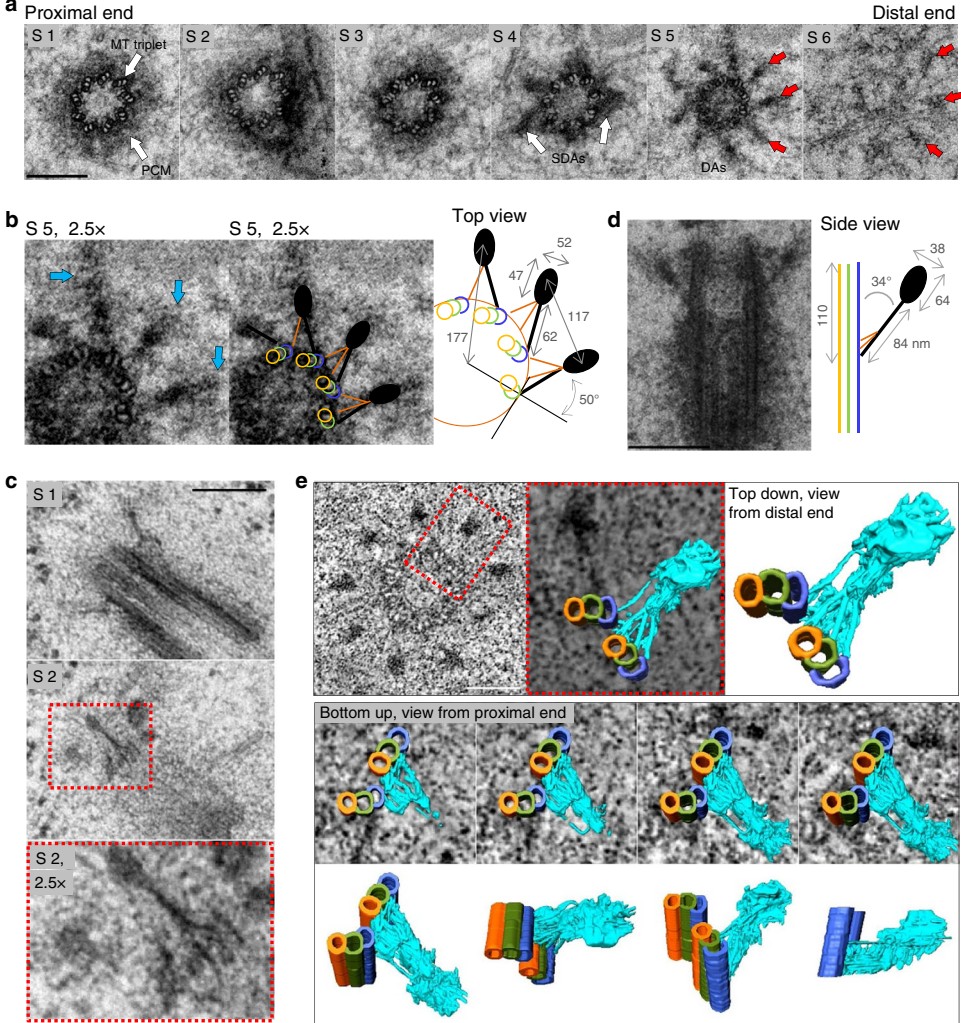

**Fig. 1** Electron microscopy characterization of distal appendages. **a** Six 80 nm serial sections (S1–6) through a mature centriole, from a HeLa cell, after chemical fixation. On the proximal end, nine microtubule (MT) triplets (S1, arrows) are surrounded by an electron dense pericentriolar material (PCM). Four subdistal appendages (SDAs) are visible in S4 (arrows). The distal appendages (DAs) are visible in S5 and S6 (arrows). **b** Enlarged detail of S5 from (**a**). A scheme illustrates major DA features and average dimensions of the DA densities in the crosscut ($n = 8$ centrioles). Blue arrows: regions of intermediate electron density surrounding a dense DA heads. The centriole A, B, and C MTs are delineated in yellow, green, and blue, respectively. The fibers at the base of the appendage and the head densities are illustrated in orange and dark grey. **c** Two 80 nm serial sections (S1 and S2) through almost longitudinally sectioned mature centriole from a HeLa cell, after high-pressure freezing and substitution. The filamentous nature of the DA base is accentuated (S2 and enlarged detail from S2). **d** An 80 nm longitudinal section through a mature centriole from a HeLa cell. Two DAs protrude at an angle of ~34 degrees. Scheme: The average dimensions of DA when sectioned longitudinally ($n = 7$ centrioles). **e** Upper row: A zero-degree tilt projection of a 120 nm-thick portion of the basal body, from a multiciliated mTEC containing DAs, used for tomography. A structural model is built for the DA marked by the red square. On the DA model, the centriole A, B, and C MTs are delineated in yellow, green and blue, respectively. A fibrous DA's base and a head density are delineated in cyan. Middle row: Four bottom-up views showing progressive slicing through the tomogram density and the DA structural model. Bottom row: DA model shown in four different orientations. The very distal end of the head density is missing. A slice-through movie of a reconstructed tomogram and a structural model built for a DA is presented in Supplementary Movie 1 and 2. Scale bars: 200 nm in (**a**, **c**, **d**) and 100 nm in (**e**)

human health. However, the appendage architecture and the mechanisms that guide their assembly and maintenance are still largely unknown. The assembly of SDAs is initiated by the recruitment of Odf2 around the centriole's MTs, followed by the recruitment of CCDC68, CCDC120, Cep170, Cep128, and Ninein[17–19]. Some SDA components such as Ninein[17] and Cep170[18] are transiently removed from SDAs from late G2 until G1. Although Odf2 remains associated with centrioles throughout the cell cycle, SDA electron microscopy (EM) densities become undetectable in mitosis[2]. The physiological significance of SDA remodeling in mitosis remains unclear.

So far, several DA proteins (DAPs) have been identified in mammalian cells and all have essential roles in ciliogenesis or ciliary-associated processes[6–9,11,12]. DA assembly is initiated by the recruitment of C2CD3, followed by CCDC41/CEP83. CCDC41 is required for the recruitment of CCDC123/Cep89 and SCLT1. SCLT1 is, in turn, needed for the recruitment of FBF1 and Cep164. Once assembled, DAs are thought to be permanent structures, contrary to SDAs. This notion was based on an observation that EM DA densities can be detected throughout interphase and on metaphase centrioles of pig kidney embryo (PE) cells[2]. However, the levels of centriole-associated Cep164 decrease before mitosis[16,20], suggesting that DAs might remodel in mitosis similar to SDAs.

If analyzed by EM, SDAs appear as robust, often striated, cone-shaped dense structures protruding from the centriole's wall[1,2] (Fig. 1a). SDAs are present in variable numbers, while DAs always appear as nine densities (Fig. 1a)[1,2,21–24]. Based solely on their appearance in two-dimensional EM analyses, DAs have been characterized as trapezoidal pinwheel-like sheets, which are rotated in the opposite direction from the centriole's MTs[22]. They are also thought to attach to one MT triplet with the smaller base of the trapezoid.

A challenge in the analysis of appendage organization arises from their small size and biochemical complexity. Recent employment of super resolution microscopy techniques has provided initial insights in the localization of SDA and DAPs[19,25–27]. However, previous studies mostly inferred the position of DAPs along the appendage ultrastructure because of the lack of high-resolution light microscopy analysis in correlation with EM. Such a correlative approach is necessary to precisely localize appendage components in relation to the centriole's MTs and EM densities, but has not yet been reported. Thus, we decided to combine the strengths of Stochastic Optical Reconstruction Microscopy (STORM, which can localize fluorescent epitopes with a resolution of ~25 nm), with 2D and 3D EM analysis (which allows for the detection of a centriole's and its appendage's ultrastructural features) to unravel the localization of key DAPs, dissect the process of DA assembly, and investigate the dynamic properties of DA components, and relate them to their ultrastructural features. Our insights greatly advance the understanding of the formation, maintenance, and architecture of these vital cellular structures.

## Results

**Distal appendages are radial protrusions**. Ultrastructural analysis of mammalian DAs is scarce. Prior EM analyses conducted on different species refers to DAs as pericentriolar satellites in jellyfish[28], alar sheets in rhesus monkey oviduct[22], or transitional fibers in human leukocytes[24] and *Chlamydomonas*[29]. These analyses provided somewhat different descriptions of DAs, which may reflect their species-specific variability. Thus, we first characterized how DA EM densities appear in human and mouse cells after using chemical fixation followed by plastic embedding and serial 80 nm sectioning[23]. In cross sectioned centrioles, DA EM

densities are found within two consecutive sections (Fig. 1a, section S5 and S6). When observed from the distal centriole end, we found that each DA is anchored to the centriole via a triangle-shaped base, comprised of filamentous densities. Each triangular base associates with one MT triplet, and with the outer, C tubule, belonging to the adjacent MT triplet (Fig. 1b). The filamentous densities fuse forming a narrower electron dense area, which continues into a wider density (hereafter referred to as the "head"). The dense heads are, additionally, surrounded by an irregularly-shaped material of intermediate density (Fig. 1b, blue arrows). The side of the triangular base which associates with the full MT triplet appears denser. This side in continuity with the head density, is turned ~50 degrees in the opposite direction from MTs (Fig. 1b), and gives a perception of a 'blade' as described in ref. [22]. The triangular and filament-like nature of the DA base was also visible in RPE-1 (Supplementary Figure 1b), and it was accentuated if high-pressure freezing followed by freeze substitution is used instead of a classic chemical fixation (Fig. 1c).

In cross sectioned centrioles (Fig. 1a, b), the centers of the heads are positioned ~177 nm away from the centriole's center and are separated by ~117 nm. If the line is drawn to connects a DA head with a centriole center, it always passes between two MT triplets. This DA organization also appears on basal bodies containing primary cilia in RPE-1 cells, and on mouse tracheal epithelial cells (mTEC) harboring motile cilia (Supplementary Figure 1a and b). In a longitudinal orientation, DAs connect to the centriole wall via a narrow density, followed by a dense head (Fig. 1d). DAs angle ~34 degrees with respect to centriole's longitudinal axis and their length measured from the base to the end of the head is ~148 nm. The bases of DAs are placed ~110 nm from centriole's distal end.

Our 2D EM analysis contradicts the proposed DA sheet-like appearance, albeit recognizing similar structural elements as previously described. Rather, in our studies, DAs appear as slightly curved, almost radial finger-like protrusions anchored to two MT triplets by a fibrous base. The finger-like appearance of EM densities and their association with two MT triplets were also apparent on oblique, 120 nm-thick sections (Supplementary Figure 1c). However, in 2D EM analysis, imaged densities are merely projections of 3D structures and the perceptual dimensions of DAs are highly sensitive to the sectioning angle, making them difficult to judge. So, to fortify our conclusions, we additionally analyzed DAs by electron tomography, which allowed us to create an accurate 3D model of the DAs (Fig. 1e and Supplementary Movie 1 and 2). This analysis revealed an intricate fibrous structure at the DA's base and confirmed its association with two adjacent MT triplets. It also revealed the finger-like appearance of DAs, corroborating our conclusions deduced from 2D EM analysis. We thus propose that such DA organization is representative for mammalian centrioles, as it was present in all cell types examined. We suggest that a robust fibrous base anchored to two MT triplets and the finger-like DA arrangement would indeed be more efficient in positioning and stabilizing relatively large DA assemblies around the centriole, when compared to organization as trapezoid-shaped sheets, linked to only one MT triplet.

**3D distribution pattern of distal appendage proteins**. Microscopy analysis of immunolabeled DAPs previously demonstrated that they form circles of various diameters around the mother centriole (Fig. 2a and[3–5,8]). But how the DAPs are arranged with respect to DA EM densities remained unanswered due to the lack of correlative high-resolution light and EM approaches. As the first step toward answering how DAPs localize around centrioles, we analyzed the organization of the core DAPs; CCDC41, SCLT1,

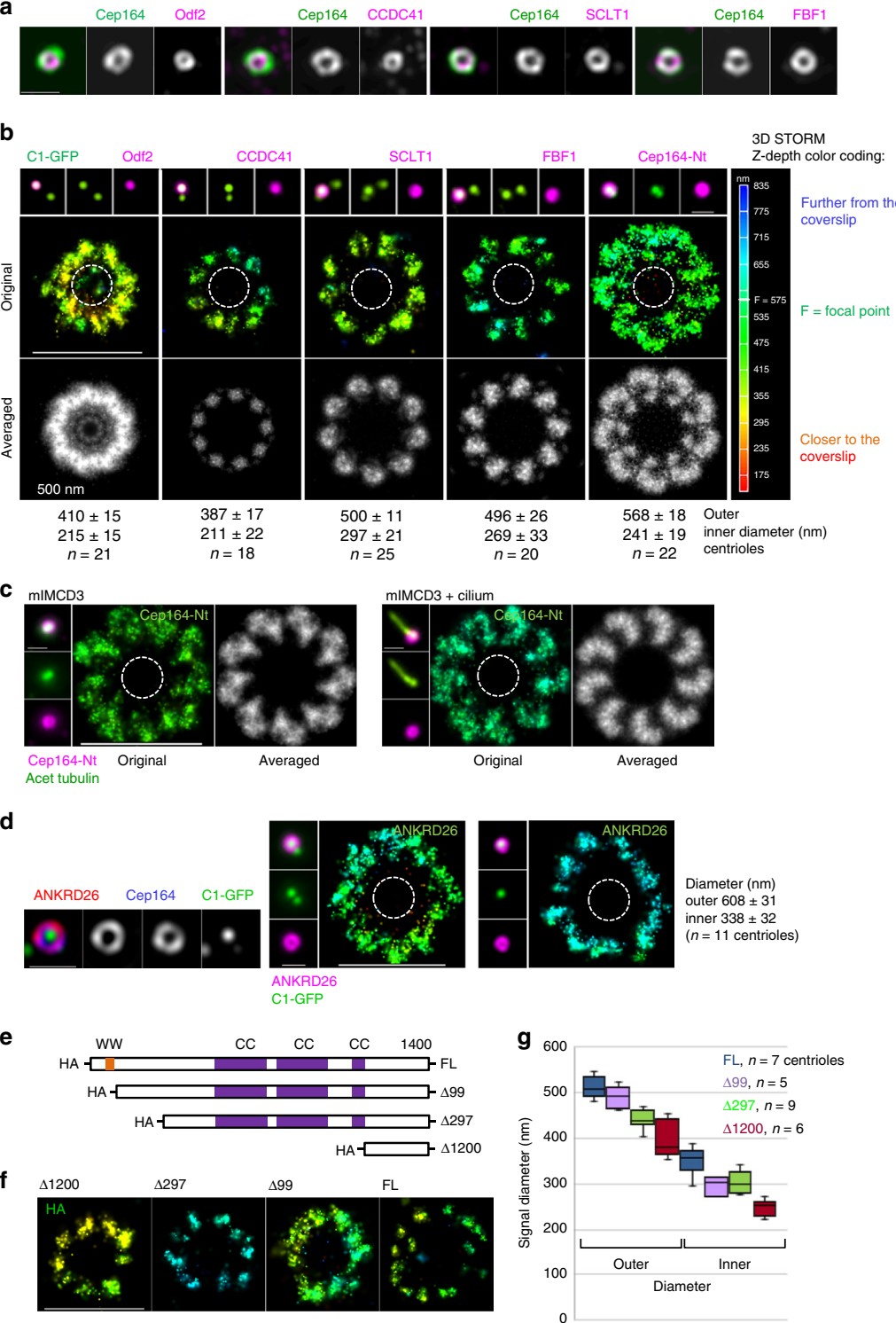

FBF1, and Cep164 by 3D STORM. We used retinal pigment epithelial (RPE-1), HeLa, and mouse inner-medullary collecting duct cells (mIMCD3). Cells expressing Centrin1-GFP (C1-GFP) were plated on coverslips and immunolabeled for individual DAPs. The centriole of interest was first imaged in a wide-field mode to select mother centrioles oriented perpendicular to the imaging plane for STORM imaging. The position of mother and daughter centriole C1-GFP signals were used to determine the mother centriole's proximal-distal orientation (Supplementary

Figure 7a). STORM revealed that each DAP exhibits a characteristic nine-fold pattern of distribution (Fig. 2b and Supplementary Figure 2). CCDC41 localized closest to the centriole MTs and was organized in nine discrete densities. These densities frequently contained a central region of lower signal intensity. SCLT1 signal also formed nine densities, but they were localized further from the centriole's center than CCDC41. Like CCDC41, SCLT1 densities often contained a central region devoid of signal. FBF1 appeared as nine densities, which were at a similar distance

**Fig. 2** 3D STORM reveals the DAP's unique nine-fold distribution pattern. **a** Two color SIM images illustrating that the DAPs, as well as the SDA protein Odf2 are organized in toroids of various diameters. **b** A comparative wide-field and 3D STORM analysis of the SDA component Odf2 and of various DAPs reveals that each protein has a unique pattern of distribution. Each STORM image is accompanied with a black and white image showing the averaged signals from all nine appendages. CCDC41, FBF1, and SCLT1 are organized in discrete signals, while Cep164 N terminus (Nt) occupies loop-like densities that touch laterally. Signals are shown from the proximal end. The Z-depth color coding scheme for STORM images is used throughout the manuscript and is illustrated to the right. **c** A loop-like organization of Cep164 N-termini in mIMCD3 cells without and with primary cilium. Elongated acetylated tubulin signal was used to detect cilia. (**d**) Characterization of ANKRD26 localization. Left: SIM reveals that ANKRD26 localizes as a toroid around older mother centriole and colocalizes with Cep164. Right: A comparative wide-field and 3D STORM analysis reveals that ANKRD26 organizes as a toroid consisting of irregularly-shaped lobules. Outer and inner diameters in (**b**, **d**) are averages ± s.d. Dashed circles in (**b**, **c**, **d**) delineate approximate position of mother centriole microtubules. **e** A full-length (FL) and three truncated Cep164 constructs were generated to determine the spatial organization of Cep164 on appendages. **f** The fragments from (**e**) were expressed in cells depleted of endogenous Cep164, labeled against HA, and analyzed by STORM. The examples of STORM images are shown. The C-terminal, Δ1200, fragment localizes closest the centriole center, indicating a radial Cep164 organization. **g** Quantification of HA STORM signals. A median line and upper and lower quartile is presented in box-and-whisker plots. Scale bars: 1 μm for all wide-field images of centrioles and 500 nm for STORM images. (This Figure is associated with Supplementary Figure 2 and 3)

from centriole centers as SCLT1 but showed more variability in size and shape than CCDC41 and SCLT1. Finally, an antibody recognizing the first 112 N-terminal residues of Cep164 revealed that Cep164 occupies a much wider area than the other DAPs. Cep164's signal was distributed in nine loop-like units, with the loop bending anticlockwise if viewed from the proximal end. The loop-like units often touched laterally, homogenizing the space between DAs (Fig. 2b and Supplementary Figure 2). Such characteristic Cep164 arrangement was preserved on both ciliated or non-ciliated mIMCD3 centrioles (Fig. 2c), and on centrioles of HeLa cells (Supplementary Figure 3a).

We additionally characterized the localization of ANKRD26 (Ankyrin repeat domain 26), a putative Cep164 binding partner (https://thebiogrid.org/), and the knockout of which perturbs ciliary function in the central nervous system in mice[30]. STORM analysis showed that in interphase, ANKRD26 colocalizes with Cep164, and forms a toroid with an inner and outer diameter of ~314 and ~578 nm, respectively (Fig. 2d). Several wide irregular lobules could usually be distinguished within a toroid. The region where ANKRD26 was detected corresponds to the outer portions of Cep164, indicating that ANKRD26 could accumulate downstream from Cep164. Indeed, Cep164 and other DAPs still localized to the centrioles in ANKRD26$^{-/-}$ cells (Supplementary Figure 3b). Detailed characterization of DAP's 3D STORM densities is presented in Supplementary Figure 2b.

Cep164 is the largest DAP we analyzed, and the localization of its N terminus showed a broader distribution pattern around centrioles compared to other DAPs. The N-terminus of Cep164 contains two tryptophan-rich domains (WW), which facilitate protein-protein interactions (Fig. 2e). This region, as an example, binds Tau tubulin kinase 2 (TTBK2)[9], an enzyme critical to the initiation of ciliogenesis[31]. Thus, broad distribution of Cep164 N-termini could act as a platform for association with other DA components. Previous studies have shown that Cep164 binds to the centriole via it's C terminal domain[20]. So, to elucidate the organization of Cep164 on DAs and to determine the position of its centriole-binding domain, we generated a series of Cep164 truncations tagged with RFP-HA (Fig. 2e), expressed them in Cep164-depleted cells, and imaged the position of their HA domains by STORM. All truncated constructs localized in a ring-shaped pattern (Fig. 2f). The shortest C-terminal fragment (Δ1200) localized in nine discrete densities, situated close to the centriole. Longer fragments were found at increasingly larger distances from the centriole, indicating that Cep164 is organized radially from its centrally positioned C-termini (Fig. 2f, g). The C-termini localized at the distances where EM head densities would be detected by EM, which prompted us to speculate that Cep164 C-termini anchor at these densities and its N-termini spread in a loop-like fashion around them.

**Correlative STORM/EM analysis of distal appendage proteins.** To unravel where STORM signals localize with respect to EM densities and centriole MTs, we employed correlative 3D STORM and EM analysis. DAPs were first imaged by STORM, post-fixed, and the same centrioles were serially sectioned and imaged by EM, all while maintaining the original vertical orientation of centrioles through embedding and sectioning. Electron micrographs of centriole sections (80 nm) were manually aligned and scaled up based on the determined shrinking coefficient, which inevitably occurs during EM sample preparation[32]. Then, the corresponding STORM image was aligned with EM micrographs (Fig. 3a and Supplementary Movie 3). CCDC41 signals aligned with the regions below EM head densities. Although filamentous bases of DAs were not preserved well in CLEM experiments (due to the lower structural preservation after immunostaining), CCDC41 most frequently occupied the space above and in between two MT triplets where the appendage bases would normally be observed by EM (Fig. 1c). FBF1 signal colocalized or near-colocalized with head densities. SCLT1 staining colocalized with the tips of the DA heads and often extended beyond them. Characteristic loop-like signals marking the N-termini of Cep164, encircled and extended beyond electron dense heads, filling the regions between DA electron-densities. Such localization of Cep164 N-termini agrees with our prediction that Cep164 C-termini co-localize with the head densities.

Determining the positions of DAPs with respect to centriole MTs and EM head densities, allowed us to superimpose STORM images and obtain their horizontal distribution map. For that purpose, CLEM micrographs were rotated to reach a maximal overlap between EM head densities (Supplementary Figure 15). Following the same rotational scheme, we rotated corresponding STORM images and superimposed them (Fig. 3b). Superimposed images revealed almost radial organization of DAPs, in agreement with EM data. To further validate our CLEM approach and confirm that DAP signals were properly aligned, we additionally performed two-protein STORM analysis. We expected that labeling and imaging two DAPs simultaneously in one channel should result in the pattern obtained by superposition of the CLEM images. Indeed, CCDC41 and SCLT1 formed nine elongated signals, in which the more centrally localized CCDC41 signal could be distinguished from SCLT1 signal, that localized further from the centriole (Fig. 3c and Supplementary Figure 4). A combination of CCDC41 and FBF1 showed a similar pattern. SCLT1 in a combination with FBF1 resulted in nine wider signals, in agreement with their expected partial overlap. Finally, the addition of CCDC41 reduced the inner diameter of Cep164 signal and diminished the appearance of empty regions within Cep164 loops. Thus, two-protein STORM yielded the expected dimensions and the localization of DAP signals,

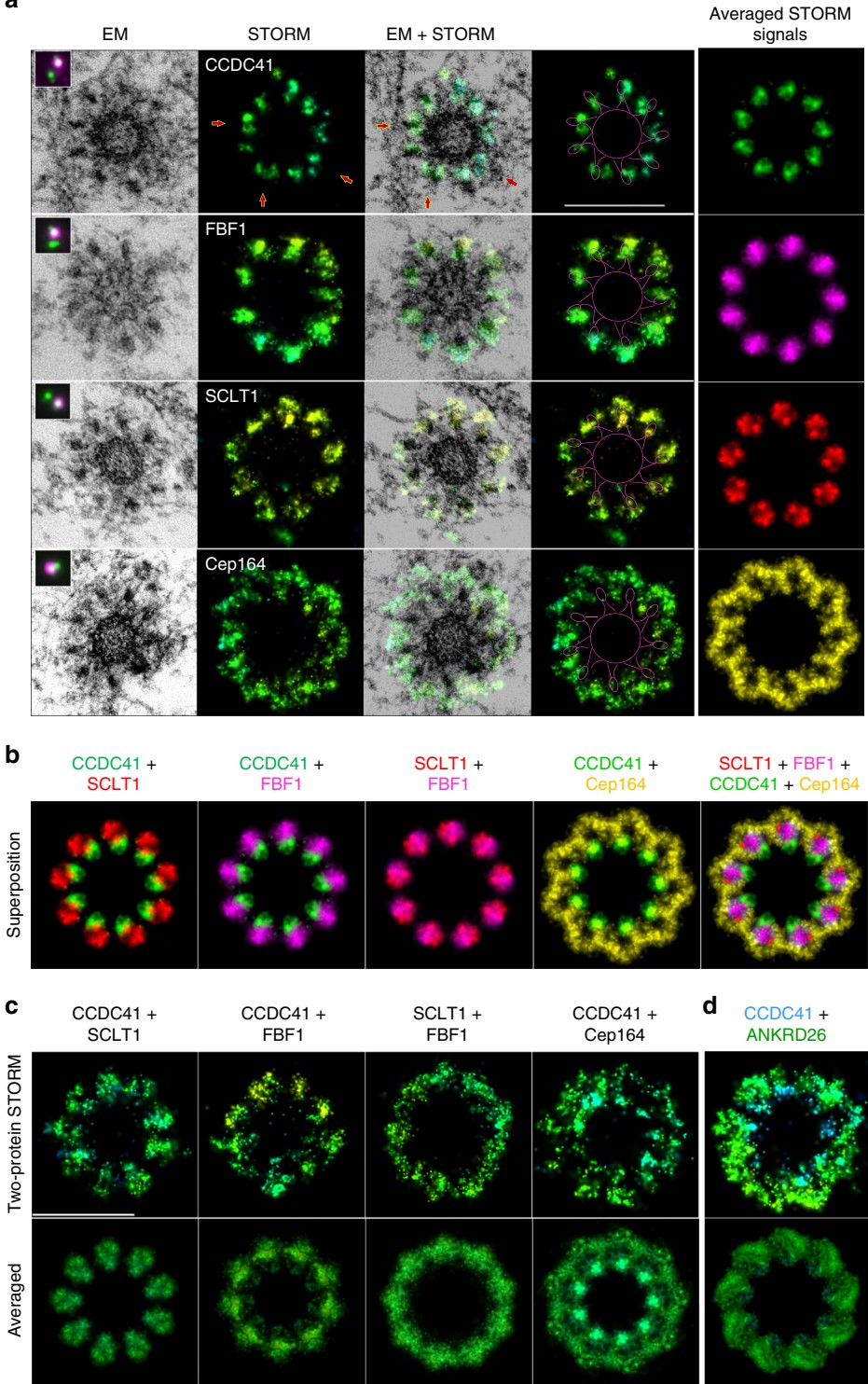

**Fig. 3** Correlative STORM/EM analysis positions DAP signals with respect to EM densities. **a** RPE-1 C1-GFP cells were immunolabeled for the indicated DAPs and imaged first in a wide-field mode (magenta in inserts), followed by 3D STORM imaging, and correlative electron microscopy analysis. The panels show one 80 nm section containing DA EM densities, the STORM image of the same centriole, merged STORM and EM image, and the STORM image with a scheme delineating the centriole and the DA head densities. The averaged STORM signal is shown in the right column. Averaged signals were pseudo colored. **b** Averaged STORM images from (**a**) were rotated and then superimposed to generate a horizontal distributional map of the DAPs. **c**, **d** Two protein STORM analysis of indicated DAPs. Cells were simultaneously labeled for two DAPs in one channel and imaged by STORM. The averaged signals from all nine appendages are shown below. More examples of two-protein STORM are shown in Supplementary Figure 4, where the average outer and inner diameter of the toroid is noted. Scale bars: 400 nm

validating our CLEM approach. We additionally preformed two protein STORM analysis of CCDC41 and ANKRD26 (Fig. 3d). More proximally situated CCDC41 signals were easily distinguished from ANKRD26 signal localized in a larger diameter and more distally, forming a wider 'platform' toward centriole distal end.

**DAP signals show a proximal-distal shift**. To determine how each DAP signal is distributed laterally, along the proximal-distal axis of the centriole, we first imaged DAP signals on horizontally oriented centrioles by STORM. CCDC41, SCLT1 and FBF1 signals measured ~52, 88, and 83 nm long foci, respectively (Fig. 4a). Cep164 N terminus was distributed in a wider (~141 nm) signal, which agrees with the thicker lateral distribution of Cep164 on perpendicularly imaged centrioles (Fig. 2c). The width of the ANKRD26 signal was ~115 nm. Multi-color SIM imaging was then used to determine the proximal-distal shift between Cep164 and other DAPs (Fig. 4b). The center of CCDC41 signal was shifted ~81 nm toward the centriole's proximal end, with respect to Cep164 signal. SCLT1 was positioned ~20 nm toward the proximal end, while FBF1 showed an ~21 nm shift toward the distal end. ANKRD26 colocalized with Cep164. The measurements obtained by multicolor SIM were further corroborated by two-protein STORM. We simultaneously labeled CCDC41 and FBF1, CCDC41 and SCLT1, or CCDC41 and ANKRD26 and measured the lateral distance between the centers of their signal. CCDC41-SCLT1 signals were separated by ~80, and CCDC41-FBF1 by ~100 nm, and CCDC41-ANKRD26 by ~80 nm (Fig. 4c), aligning with our SIM measurements (Fig. 4b). Finally, using the horizontal and vertical distributions of individual DAPs and their relative positions with respect to the EM densities, we created a 3D model for DAPs distribution, as presented in Fig. 4d.

The distribution of DAPs revealed in our experiments is in several aspects different from previously suggested distribution[27]. The most notable distinction is in distribution of Cep164 and FBF1 N-termini with respect to centriole MT triplets and EM densities, which probably originates from the use of different methodological approaches. Our correlative STORM and EM analysis in combination with genetics and two-protein STORM analysis, allowed us to unambiguously determine the localization and organization of DAP domains. Hence, we elucidated a different distribution of Cep164, where its C-termini colocalize with DA electron dense heads, while its WW-rich N-termini extend into the regions between DA's electron densities and into the regions beyond electron-dense heads, homogenizing the space around the centriole. We also precisely localized N-terminal FBF1 domains to the regions above DA head densities (Fig. 4d).

**Appendages form gradually on maturing mother centrioles**. The localization of the core DAPs along EM densities determined by CLEM follows the proposed recruitment hierarchy[8], in which proteins that localized closer to the centriole wall such as CCDC41 are needed for the accumulation of outer DAPs such as FBF1 and Cep164. This would suggest that during appendage formation on maturing centrioles, the inner DAPs accumulate before the outer DAPs. To address the timing of DA assembly with respect to their second cell cycle, we quantified when various DAPs first associate with the younger mother centrioles (Fig. 5a and Supplementary Figure 5a). The quantification showed that, CCDC41 indeed accumulates first and was detectable on maturing centrioles by G2, often before DNA was visibly condensed (Fig. 5a, b). CCDC41 continued its accumulation through early mitosis, reaching its highest centriolar levels in metaphase. SCLT1, the next DAP on the recruitment hierarchy, accumulated around prophase, while FBF1 and Cep164, the two outer DAPs,

were largely absent from maturing centrioles until telophase (Fig. 5a). The analysis of centrosome-associated ANKRD26 levels was somewhat hampered due to the nonspecific scattered immunofluorescence signal associated with both centrioles during mitosis (Supplementary Figure 5b). Nevertheless, we determined that ANKRD26 was not detectable on maturing centrioles in prophase, while it clearly accumulated to the young centrioles after metaphase, thus, slightly earlier than Cep164 and FBF1. The accumulation of Cep164, and FBF1 continued throughout early G1 (within ~1 h after mitosis). Odf2, the essential SDA protein, gradually accumulated on the maturing centrioles through mitosis and early G1 (Fig. 5a and Supplementary Figure 5a).

Next, we used STORM to reveal how DAs form in high resolution. From its first appearance on maturing centrioles in G2, CCDC41 localized in discrete densities, which were spaced at the distances typically found on fully mature centrioles (Fig. 5b). This confirms that CCDC41 can organize in nine-fold fashion even in the absence of outer DAPs. SCLT1, which started to accumulate in prophase, was present in variable densities until late mitosis (Fig. 5c). Cep164 accumulation on younger centrioles was still incomplete in late mitosis and the STORM signal consisted of sporadic irregularly shaped densities (Fig. 5d). In anaphase, ANKRD26 formed a smaller toroid (Fig. 5e) than the one measured in interphase (Fig. 2d). Thus, although inner DA components CCDC41 and SCLT1 start accumulating during centriole's second G2 phase, in what could be a nine-fold DA scaffold, DAs are not molecularly complete before early G1.

**Distal appendages remodel before mitosis**. We next explored whether DAs, once formed on mature centrioles, are stable structures or are modified in mitosis, as suggested by the reduction of centriole-associated Cep164[16,20]. First, we quantified the levels of DAPs associated with the older parental centriole and found that they behave differently (Fig. 6a and Supplementary Figure 5a). CCDC41 was present throughout the cell cycle and even transiently accumulated on the centrioles from prophase to metaphase. SCLT1 levels remained relatively constant, although we regularly detected a transient ~20% loss of SCLT1 in prophase. FBF1 levels were on average reduced two-fold by metaphase, but careful analysis has revealed that ~50% of older mother centrioles maintained near-interphase levels, while the other ~50% lost some or most of the signal (Fig. 6b). STORM analysis of the remaining FBF1 signals, in early mitosis, showed a correlation between decreased FBF1 intensity and a reduced number of FBF1 STORM densities (Fig. 6c). The reason for this bipartite FBF1 behavior in a cell population remains unclear to us. Consistent with previous reports[16,20], the levels of centriole-associated Cep164 decreased in G2 and, by prometaphase, Cep164 was largely removed from centrioles. It gradually re-accumulated at the end of mitosis (Fig. 6a). STORM imaging of residual Cep164 signal in G2 and early mitosis revealed a variable number of weak Cep164 signals closer to the centrioles and a loss of the outer Cep164 layers and its loop-like organization (Fig. 6d). In addition, its binding partner TTBK2 was also lost from centrioles before mitosis (Supplementary Figure 6a). Another outer DAP ANKRD26, was undetectable on older mother centrioles by early prophase (Supplementary Figure 5b) and re-accumulated during late mitosis (see also the older mother centriole in Fig. 5e).

Consistent with reduced levels of DAPs, the DA EM densities were also less apparent and sometimes altogether undetectable around older mother centrioles in prophase and prometaphase (Fig. 6e). Quantification of remaining densities demonstrated that the average DA head density, as well as their overall size, was significantly reduced (Fig. 6f). The distances between adjacent heads appeared more variable, indicating that DAs may be less

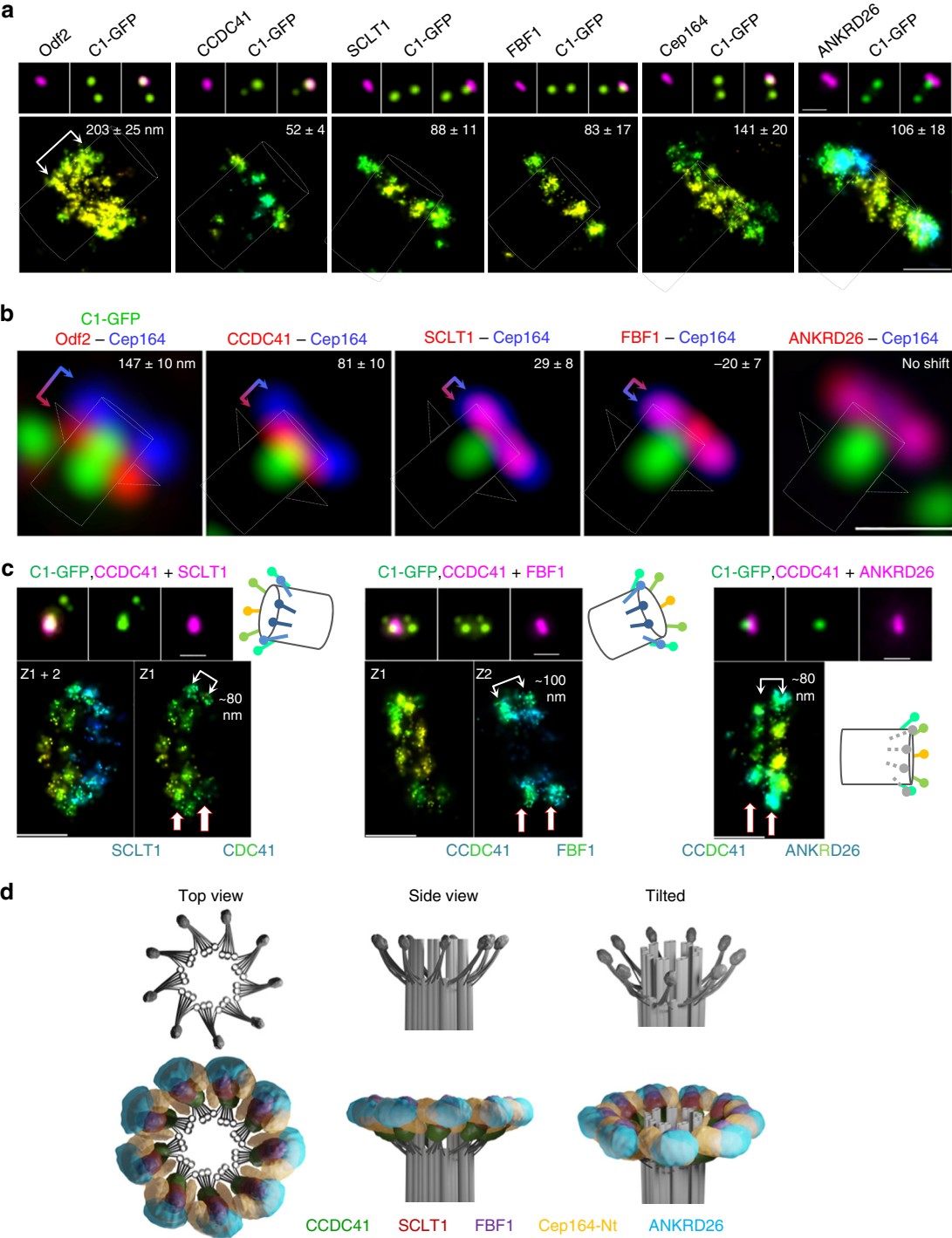

**Fig. 4** Longitudinal distribution of distal appendage proteins. **a** A comparative wide-field and 3D STORM analysis of the SDA component Odf2 and various DAPs on longitudinally-oriented older mother centrioles in interphase. The average ± s.d. thickness of the signals is noted. $n = 11$ centrioles for Odf2, 10 for CCDC41, 22 for SCLT1, 16 for FBF1, 26 for Cep164, and 10 for ANKRD26. **b** Two-color SIM images illustrating the lateral shift between the DAPs. The average ± s.d. distance between centers of each signal's intensity peaks is indicated. CCDC41 signal localizes most distally, while SCLT1 almost colocalizes with Cep164. FBF1 signal appears slightly shifter towards the distal end. ANKRD26 and Cep164 colocalize. $n = 8$ centrioles for Odf2, 11 for CCDC41, 17 for SCLT1, 15 for FBF1 and 10 for ANKRD26. **c** Two-protein STORM analysis of longitudinally-oriented centrioles showing that CCDC41 and FBF1 are laterally shifted for ~100 nm, and CCDC41 and SCLT1 and CCDC41 and ANKRD26 for ~80 nm, corroborating values obtained in (**b**) by SIM. **d** A 3D model illustrating the organization of DAs electron densities, and the localization of the DAPs with respect to the electron densities. Scale bars: 1 μm for all wide-field images of centrioles and 200 nm for STORM images

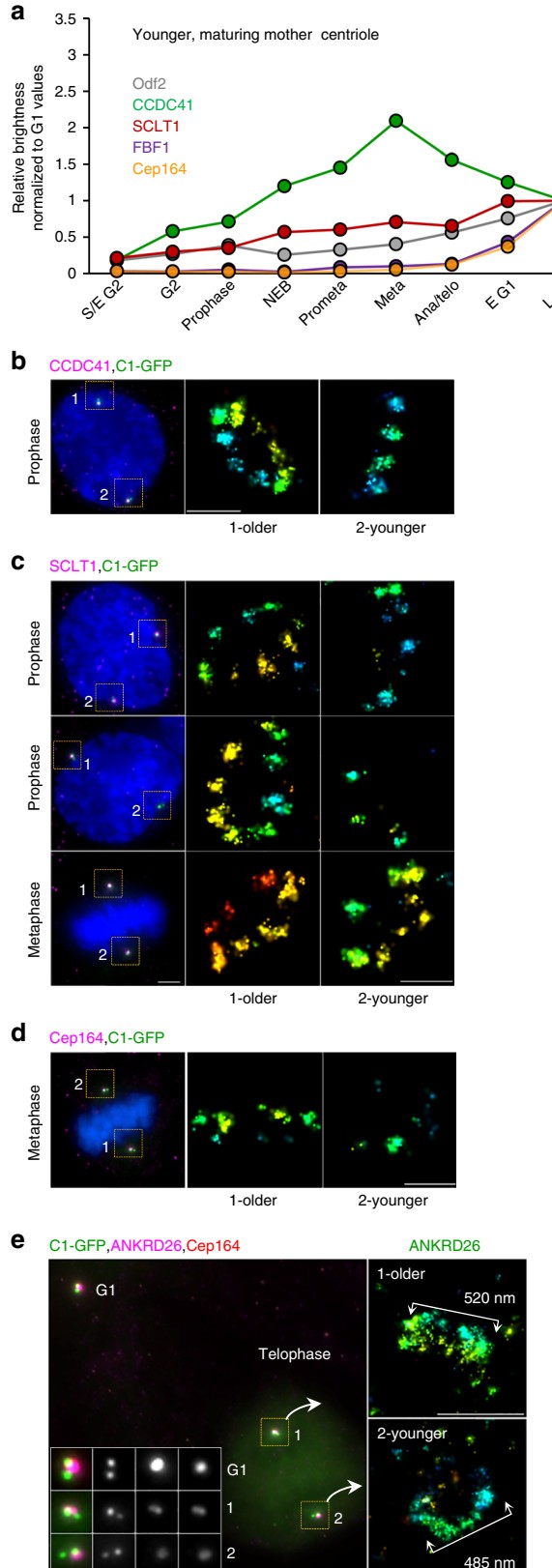

**Fig. 5** Distal appendage assembly on younger mother centrioles. **a** Average levels of the DAPs and Odf2 associated with younger mother centrioles in RPE-1 during the cell cycle. n = 217 cells for Cep164, 135 for CCDC41, 155 for SCLT1, 176 for FBF1, 244 for Odf2. Box-and-whisker plots of the same dataset are presented in Supplementary Figure 5. Representative results from a single dataset; the quantification was performed several times with similar results (3x for Cep164 and FBF1, 2x for SCLT1, CCDC41 and Odf2). **b** In prophase, CCDC41 is localized in discrete foci on both the young and old mother centriole. **c** SCLT1 signal can be detected on maturing centriole from prophase to metaphase, but it is still not organized in nine densities as visible on older mother centrioles. **d** The levels of Cep164 are low in metaphase and the signal is not structured. Cep164 associated with older mother centriole is less abundant and unstructured than in interphase (compare with Fig. 2b and d). **e** In telophase, ANKRD26 is associated with the both mother centrioles but is organized in a smaller toroid than in interphase (compare with Fig. 1h). Scale bars: 1 μm for wide-field images; 200 nm for STORM images in (**b**, **c**, **d**) and 500 nm in (**e**). The source data underlying Figure 5a is provided as a Source Data file

protein STORM analysis of CCDC41 and SCLT1 in prometaphase showed both proteins maintained nine-fold organization, like in interphase (Fig. 6g, and the older mother centrioles in Fig. 5b, c). Based on this data, we speculate that the transient displacement of Cep164, FBF1, ANKRD26, TTBK2, and possibly some still undefined DA components contributes to the diminished detectability of DAs by EM in mitosis.

The physiological significance of pre-mitotic DA reorganization is unclear. It has been shown that RPE-1 cells in G1 which inherit the older mother centriole tend to grow primary cilium ahead of their sisters that inherited the younger mother centriole[33]. Thus, DA remodeling could serve to "reduce the age gap" between two mother centrioles during mitosis and early G1. Removing outer functional components which mediate ciliogenesis or signaling from older centrioles before mitosis and gradually restoring them in early G1 (Fig. 6i), could be a mechanism for balancing centrosome-associated functions between two sister cells in early G1, before younger mother centriole's DAs are fully assembled. Pre-mitotic loss of outer DAPs could also assure timely cilia reabsorption.

It is also noteworthy that the inner DAP component CCDC41, and to a large extent SCLT1, which gradually accumulate on the younger centrioles during mitosis, remain continuously associated with mature centrioles and even retain a nine-fold organization. We speculate that the inner DAPs such as CCDC41 and SCLT1 might play a structural role in assembling the appendage's scaffold. Maintaining the scaffold, while modulating the levels of only peripheral, more functional DA components by cell cycle regulators, would allow functional adaptation of DAs to various cell cycle requirements, and a quicker restoration of the DA's interphase functions after mitosis.

**Plk1 and Aurora A modulate Cep164 dynamics before mitosis.** Pre-mitotic DA reorganization appears to be a complex process, which probably affects other more outer DAP components in addition to the ones analyzed here. Cytosolic levels of Cep164 do not change during mitosis[6], so it is plausible that its dissociation from the centrioles, before mitosis, is regulated via posttranslational modifications. Along those lines, phosphorylation of some SDA and DAPs changes during the cell cycle[6,15,18]. DA remodeling occurs prior to mitotic centrosome maturation, which is regulated by mitotic kinases such as Plk1 (Polo-like kinase 1), and Aurora A. Plk1 and Aurora A, together with Nek2 kinase have been implicated in cilia reabsorption (reviewed in ref. [10]). This

structured, albeit still organized in nine-fold fashion, during that time. In agreement, correlative STORM/EM analysis of the inner DAP CCDC41 in prophase (Fig. 6h) showed that it remained organized in nine discrete densities, although the SDA and the DA head densities were no longer detectable. Further, two-

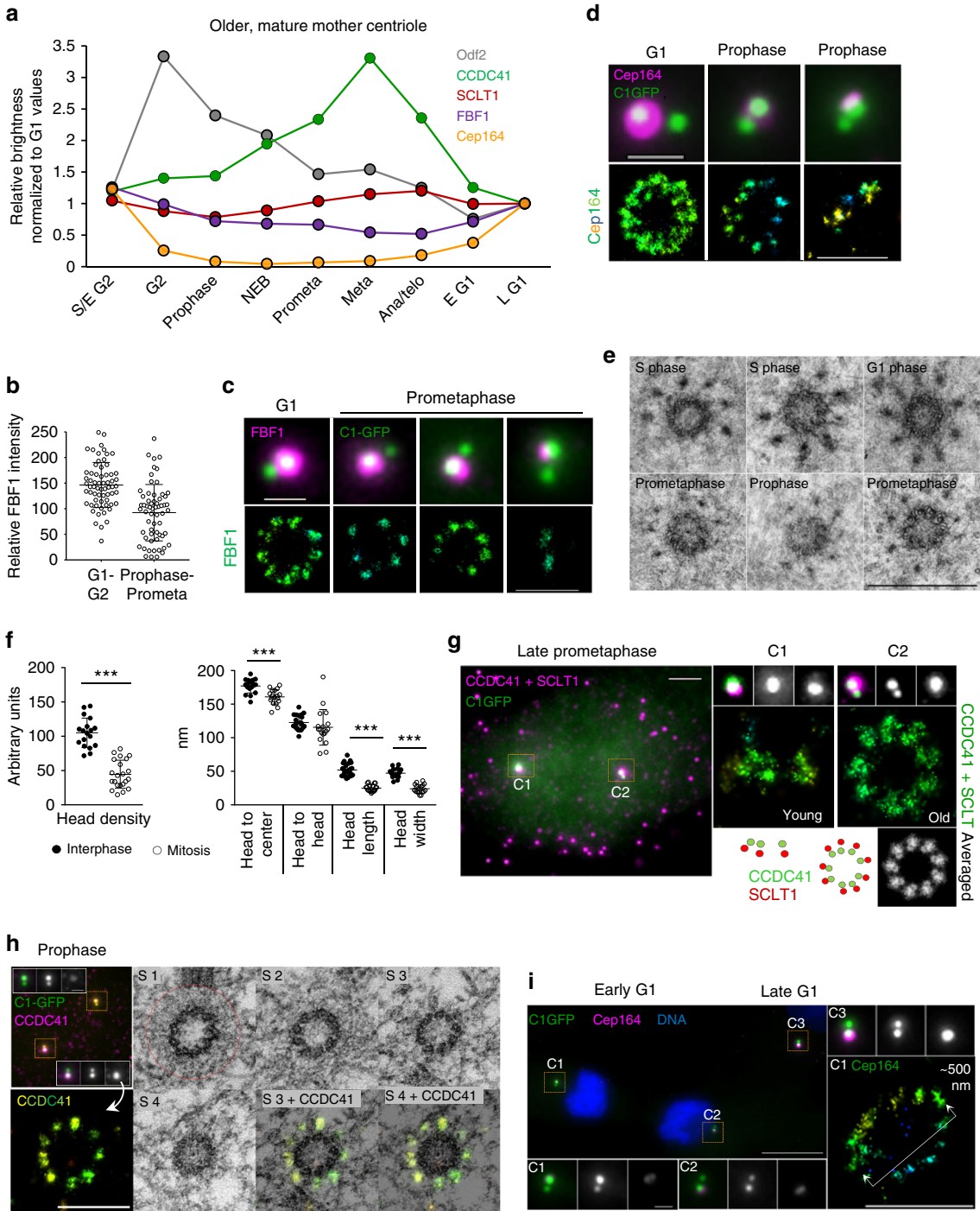

prompted us to investigate whether the association of Cep164 is dependent on Plk1 or Aurora A. Cells were treated with BI2536 (BI) to inhibit Plk1 or MLN8237 (MLN) for Aurora A inhibition, incubated for 2–4 h, and immunolabeled. The levels of centriole-associated DAPs were assessed from prophase to prometaphase. Inhibition of Aurora A did not significantly change the dynamics of DAPs in HeLa cells (Fig. 7a, b). However, inhibition of Plk1, in some cells partially and in some cases completely, prevented Cep164 removal from older centrioles. We observed similar results in in mIMCD3 cells (Fig. 7c, d). In mIMCD3, Aurora A inhibition also prevented the loss of Cep164, albeit to a lesser

degree than Plk1. Plk1 inhibition also reduced the loss of TTBK2 and ANKRD26 before mitosis (Supplementary Figure 6). To assess a potential relationship between Cep164 removal and cilia reabsorption, we used mIMCD3 cells, which ciliate in the presence of serum and normally absorb cilia in G2, before cells exhibit visible signs of DNA condensation. We treated mIMCD3 with Plk1 and Aurora A inhibitors and analyzed the dynamics of cilia reabsorption. We identified a cilium by the presence of an elongated acetylated tubulin signal associated with Cep164 (Fig. 7e). In control cells, all cilia reabsorbed before prophase. In contrast, ~50 or ~20% of prophase cells maintained their cilium

Fig. 6 Distal appendages of mature centrioles remodel before mitosis. **a** Average levels of the DAPs and Odf2 associated with older mother centrioles in cycling RPE-1 during the cell cycle. $n = 217$ cells for Cep164, 135 for CCDC41, 155 for SCLT1, 176 for FBF1, 244 for Odf2. Box-and-whisker plots of the same dataset are presented in Supplementary Figure 5. Representative results from a single dataset; the quantification was performed several times with similar results (3x for Cep164 and FBF1, 2x for SCLT1, CCDC41 and Odf2). **b** Intensity of FBF1 signals is variable on older mother centrioles in mitosis. A median line and upper and lower quartile are marked in dot-plots, $n = 63/61$ for interphase/mitotic centrioles. **c** Decreased FBF1 intensity in prometaphase correlates with reduced number of STORM foci. **d** The typical interphase loop-like organization of Cep164 is no longer detectable on older mother centrioles in mitosis. **e** Comparison of the DA EM densities of interphase and mitotic cells. One 80 nm section is shown for each centriole. **f** Quantification of DA's EM densities from interphase and mitotic centrioles. A median line and upper and lower quartile are marked in dot-plots, $n = 5$ centrioles for mitotic and $n = 7$ for interphase samples.***$P < 0.001$. **g** Left panel: wide-field image of prometaphase RPE-1 cell immunolabeled for CCDC41 and SCLT1 in one color (magenta). The same cell was analyzed by STORM (right panel). **h** Correlative STORM/EM analysis of older mother centriole's CCDC41 in prophase. Left: wide-field image of a prophase RPE-1 cell immunolabeled for CCDC41 and analyzed by STORM. Each mother centriole (containing brighter C1-GFP) is associated with a daughter centriole (adjacent dimmer C1-GFP). Right: Correlative EM analysis of the same older mother centriole from left panel. CCDC41 is localized to nine discrete signals, but DAs and SDAs densities are no longer detectable. The PCM is outlined by red circle. This panel is associated with Supplementary Figure 2. **i** Two sister cells in early G1, at the stage of Cep164 re-accumulation to the mother centrioles (C1 and C2). STORM shows the lack of Cep164 arrangement typical for late G1 and S phase (compare to Fig. 2). Scale bars: wide-field images of centrioles, 1 μm; wide-field image in (**g**) 2 μm; STORM images, 500 nm in (**d**, **c**, **i**); 200 nm in (**g**); 400 nm in (**h**); 500 nm in (**e**). The source data underlying Figure 6a is provided as a Source Data file

after Plk1 or Aurora A inhibition, respectively. Such cilia were retained until late prophase but, although Cep164 remained associated with centrioles, the ciliary marker become absent soon after nuclear envelope breakdown (Fig. 7e). We concluded that, although cilia reabsorption and DA remodeling both occur concomitantly in G2, the removal of Cep164 from centrioles is not necessary but may facilitate cilia reabsorption. Nevertheless, the observed delay in cilia reabsorption after Plk1 inhibition could stem from the lack of Plk1-dependent activation of tubulin acetylase, shown to be needed for cilia reabsorption in G2[34,35]. Thus, although the mechanisms that lead to the pre-mitotic removal of DAPs remain unclear, our studies point to the direct or indirect involvement of mitotic kinases.

## Discussion

This study demonstrates the power of correlative imaging approaches in revealing architectural and dynamic properties of complex cellular structures with a nanoscale resolution. In this work, we identify novel architectural features of distal appendages and develop a precise 3D localization map for their components, revealing a radial organization of DAs. We next ascertained the timing of their initial formation and unravel their dynamic nature during the cell cycle. We further detail, in nanoscale resolution, the early phases of DA assembly, which demonstrated that DA formation occurs gradually through accumulation of the inner components CCDC41 and SCLT1 in G2. This process is followed by the accumulation of the outer components such as Cep164, FBF1 and ANKRD26, in late mitosis and during early G1. Finally, we show that DAs of mature centrioles undergo a dramatic structural reorganization before mitosis. We dissect their ultrastructural changes during that time and show that the DA's outer components are temporarily lost from centrioles, while inner components remain continuously associated and maintain a nine-fold organization. Pre-mitotic reorganization of SDAs has been previously established, but DAs have been traditionally viewed as permanent centriolar structures of mature centrioles. Our study reveals that the two types of appendages bear more similarities than previously thought and that both dramatically change before mitosis. Our unearthing of the dynamic nature of outer DAPs such as FBF1 and particularly Cep164 and ANKRD26 suggests that DAs could play a role beyond their well-established role in ciliogenesis. Cep164 and ANKRD26 both contain protein-protein interaction rich domains on their N-termini (WW domains and ankyrin repeats, respectively). Both proteins organize in a wide and almost platform-like arrangement around and above centriole distal end. Such platform composed of highly interactive proteins could serve as docking site for

various signaling molecules and, hence, participate in signaling events. It is possible that some of appendage-based signaling cascades needs to be attenuated in mitosis through a rapid disassembly of outer DA layers. Although human cells in culture proliferate in the absence of DAs, the number of pathologies associated with improper signaling due to various mutations and deficiencies in DAPs is growing[30,36–38]. A systematic genetic manipulation of DA components will be needed to dissect the mechanisms and meaning of pre-mitotic DA remodeling.

## Methods

**Cell cultures and drug treatment**. Retinal pigment epithelial RPE-1[39] and HeLa[40], constitutively expressed Centrin1-GFP (C1-GFP). Mouse inner-medullary collecting duct cells mIMCD3 were a gift from Dr. C Westlake (NCI/CCR-Frederick). The cultures were grown in DMEM (Invitrogen) supplemented with 10% fetal bovine serum (FBS) and 1% Penicillin Streptomycin (PS), at 37 °C, in a humified environment with 5% $CO_2$. RPE-1 were starved in medium containing no serum for 24 h to induce ciliation. For immunofluorescence and imaging, cells were plated on round 25 mm, 1.5 high precision cover glasses (Warner Instruments). Cells were treated with 200 nM BI2536 inhibitor (BI; Selleckchem) to inhibit Plk1 activity and with 250 nm MLN8237 (MLN; Selleckchem) to inhibit Aurora A.

**mTEC cultures**. Cultures were established as previously described[41,42]. Briefly, C57BL/6 mice were sacrificed at 2–4 months of age, trachea were excised, opened longitudinally to expose the lumen, and placed in 1.5 mg/mL Pronase E in DMEM/F12 medium (Life Technologies) at 4 °C overnight. Tracheal epithelial cells were dislodged by gentle agitation and collected in DMEM/F12 with 10% FBS. After centrifugation, cells were treated with 0.5 mg/mL DNase I for 5 min on ice and centrifuged at 4 °C for 10 min at 400 g. Cells were resuspended in DMEM/F12 with 10% FBS and plated in a tissue culture dish for 5 h at 37 °C with 5% $CO_2$ to adhere contaminating fibroblasts. Non-adhered cells were then collected, concentrated by centrifugation, resuspended in an appropriate volume of mTEC-Plus medium (described in ref. [42]), and seeded onto Transwell-Clear permeable filter supports (Corning). Air-liquid interface (ALI) was established 2 days after cells reached confluence by feeding mTEC-Serum-Free medium[42] only in the lower chamber. Cells were cultured at 37 °C with 5% $CO_2$, and media replaced every 2 days, and fixed on the indicated days. All chemicals were obtained from Sigma Aldrich unless otherwise indicated. Media were supplemented with 100 U/mL penicillin, 100 mg/mL streptomycin, and 0.25 mg/mL Fungizone (all obtained from Life Technologies).

**Creation of ANKRD26$^{-/-}$ cells**. CRISPR/Cas9 was used to target ANKRD26 in Tet-On Myc-Plk1 hTERT RPE1 cells[43]. An sgRNA targeting ANKRD26 (5′-ACT AAGCCGTCCATTATACT-3′) was cloned into a lentiGuide-puro vector (52963; Addgene). HEK293FT cells were transfected with the plasmid along with lentiviral packaging plasmids psPAX2 and pMD2.G (12260 and 12259; Addgene). Virus was harvested 48 h after transfection and filtered through a 0.45 μm syringe filter (Denville). Cells were transduced with 1 mL of filtered virus supplemented with 10 μg/mL of polybrene (MilliporeSigma). Cells were selected with 3 μg/mL puromycin (MilliporeSigma) and then limiting dilution was used to obtain monoclonal lines. ANKRD26 knock out was validated in monoclonal lines via western blot (GeneTex; GTX128255).

**Immunofluorescence**. Cells were fixed in 1.5% formaldehyde and post fixed in 100% methanol at −20 °C for 1–4 min, washed in 1×PBS, blocked in IF buffer (1%

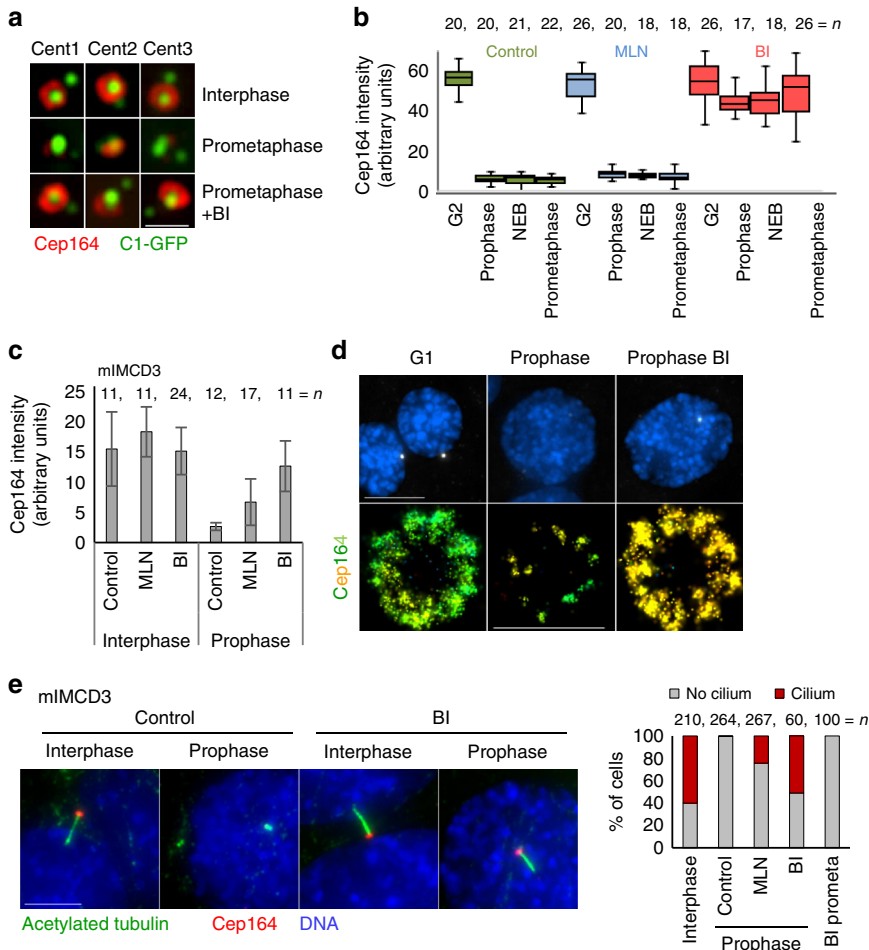

**Fig. 7** Plk1 and Aurora A modulate pre-mitotic DAP dynamics. **a** Examples of older mother centrioles from HeLa cells under different experimental conditions. Inhibition of Plk1 for 4 h (BI) prevents the loss of Cep164 from centrioles in early mitosis. **b** Quantification of Cep164 associated with older mother centrioles in HeLa cells after Plk1 (BI) and Aurora A (MLN) inhibition. A median line and upper and lower quartile is presented in box-and-whisker plot. **c** BI and MLN were added to logarithmically growing mIMCD3 cells. Two h later, cells were stained for Cep164 and quantified for the presence of Cep164 on older mother centrioles. **d** Cells were treated as in (**b**) The histogram presents averages ± s. d. STORM analysis of Cep164 distribution on older mother centrioles in mIMCD3 cells. The loss of Cep164 from older mother centrioles in prophase is diminished by Plk1 inhibition. **e** The effect of Plk1 and Aurora A inhibition on cilia reabsorption. BI and MLN were added to logarithmically growing mIMCD3 cells. Two h later, cells were stained for Cep164 and acetylated tubulin (used as a marker for cilia). Plk1 inhibition and to some extent Aurora A inhibition delay cilia reabsorption, which in control cells occurs before prophase. However, after nuclear envelope breakdown cilia reabsorbed and by prometaphase were no longer detectable. The number of cells is from three independent experiments. Scale bar: 10 μm for wide-field images; 500 nm for STORM image in (**b**)

BSA, 0.05% Tween-20, in 1×PBS) for 15 min, and incubated with primary antibody diluted in IF buffer at 37 °C for 1 h or 4 °C overnight. After washing, cells were incubated with secondary antibody, diluted IF buffer, and incubated at 37 °C for 1 h. To visualize DNA, cells were stained with a 1:2000 dilution of Hoechst (Invitrogen 33342) in PBS. The following primary antibodies were used: Cep164 rabbit (Proteintech; 22227-1-AP, recognizing aa: 1–112) at 1:1000, FBF1 rabbit (Sigma; HPA036561, recognizing aa: 39–115) at 1:100, SCLT-1 rabbit (Sigma; HPA036561, recognizing aa:144–222) at 1:50, CCDC41/Cep83 rabbit (Sigma; HPA038161, recognizing aa: 578–677) at 1:200, ODF-2 rabbit (Proteintech; 12058-1-AP, recognizing aa: 39–181) at 1:500, TTKB2 (Sigma; HPA018113) at 1:500, ANKRD26 (GeneTex; GTX128255) at 1:1000, acetylated tubulin mouse (Sigma; T7451) at 1:10000. CF568, CF647, or AF488 conjugated secondary antibodies (Biotium; CF647 anti-mouse 20042, CF647 anti-rabbit 20045, CF568 anti-rabbit 20099, anti-mouse 20109, Invitrogen; AF488 anti-mouse A11019, AF488 anti-rabbit A11034) were used at 1:800 dilution to label primary antibodies. In addition, Cep164 was directly conjugated with CF647 using a commercial antibody labeling kit (Biotium; 92259) and used at 1:250 dilution.

**Wide-field fluorescence microscopy**. Wide-field images were acquired using a Nikon Eclipse Ti inverted microscope, equipped with a 64 μm pixel CoolSNAP HQ[2] camera (Photometrics) and Intensilight C-HGFIE illuminator, using 100× NA 1.42 Plan Apo objective with 1.5× magnifying tube lens. Two hundred nanometers-thick Z-sections spanning entire cell or entire centrosome, as needed, were

acquired. ImageJ/Fiji (National Institute of Health, Bethesda, MD), and NIS-Elements software package was used to make maximal intensity projections and to assemble image panels. AutoQuant X3 software (MediaCybernetics) was used for deconvolution.

**Fluorescence intensity measurement**. Florescence intensity of centrosome-associated proteins was performed using ImageJ/Fiji. The cells in various cell cycle and mitosis stages were identified by DNA morphology and the centriole number. The position of the centrosomes/cilia was identified based on C1-GFP signals or by staining with acetylated tubulin. Integrated density of centrosome-associated proteins was measured within a defined area of a constant size encircling the centrosome from the summed intensity projections of Z stacks. Intensity of the background in a near proximity of each centrosome was subtracted from the signal intensity. At least 20 centrosomes were measured for each condition/cell cycle stage.

**Structured illumination microscopy (SIM)**. SIM was performed on N-SIM, Nikon Inc., equipped with 405, 488, 561, and 640 nm excitation lasers, Apo TIRF 100x NA 1.49 Plan Apo oil objective, and back-illuminated 16 μm pixel EMCCD camera (Andor, DU897). Hundred nanometers-thick Z Sections were acquired in 3D SIM mode generating 15 images per plane (5 phases, 3 angles) as a raw image, which was reconstructed to generate a super-resolution image. XYZ shift in multicolor images was corrected using 100 nm tetra-spectral fluorescent spheres

(TetraSpeck beads, Invitrogen) included in the mounting medium. For surface rendering, 3D visualization of the centrosomes, and various measurements, we used NIS-elements software package and ImageJ/Fiji. Wide-field images of centrioles presented in inserts were scaled 3× with bicubic interpolation.

**Stochastic optical reconstruction microscopy (STORM)**. Cells plated on coverslips were fixed and labeled with primary antibodies. CF647-conjugated FAB2 antibodies (Biotium) were used at 1:800 dilutions to label primary antibodies. Before STORM imaging, samples were layered with 100 nm tetra-spectral fluorescent spheres (Invitrogen), which served as fiducial markers. Coverslips were mounted in Attofluor Cell chambers (Thermofisher) and immersed in imaging buffer (25 mM β-mercaptoethylamine, 0.5 mg/mL glucose oxidase, 67 μg/mL catalase, 10% dextrose, in 100 mM Tris at pH 8.0). 3D STORM imaging was performed on a Nikon N-STORM4.0 system using Eclipse Ti inverted microscope, Apo TIRF 100X SA NA 1.49 Plan Apo oil objective, 405, 561,488, and 647 nm excitation laser launch and a back-illuminated EMCCD camera (Andor, DU897). The 647 nm laser line (150 mW out of the fiber and ~90 mW before the objective lens) was used to promote fluorophore blinking. The 405 nm laser was used to reactivate fluorophores throughout the imaging process. The 561 nm laser was used to record the signals of fiducial markers. Fifteen thousand to 30,000-time points were acquired at a 50 Hz frame rate each 16–20 ms. NIS Elements (Nikon) was used to analyze and present the data. Each STORM analysis was repeated multiple times, and multiple images were acquired for each experiment. Prior to STORM imaging, the position of the Centrin1-GFP-labeled centrioles and of the CF647-labeled target protein was recorded in wide-field mode. 3D STORM data is presented as a projection of the entire 3D volume. The original Z color coding scheme was used to assess and to illustrate the tilt of the centrioles with respect to the coverslip. The color coding from red (close to the coverslip) to blue (further from the coverslip) presents the calibrated Z range (Supplementary Figure 8). When needed, STORM images were overlaid with the outline of appendages to illustrate centriole orientation. Wide-field images of centrioles analyzed by STORM were scaled 3× with bicubic interpolation for presentation. The diameters and the sizes of STORM signals were measured as illustrated in Supplementary Figure 8.

**Resolution in 3D STORM experiments**. To determine the resolution of 3D STORM analysis, we used immunolabeled cytosolic microtubules (Supplementary Figure 9). Cytoplasmic microtubules are polymers of alpha and beta tubulin and have relatively constant size of ~23 nm when analyzed by EM. For 3D STORM analysis, cells were fixed cells in 1.5% glutaraldehyde in PBS for 4 min, permealized cells with 0.05% of Triton-x100 in PBS for 1 min, and labeled by primary antibody against alpha tubulin and the secondary CF647 F(ab')2 secondary antibodies (Biotium). The average outer diameter of 3D STORM signals measured at multiple sites of the sample was 52.8 ± 5.3 nm. This value consistent with the average diameter of the MTS (23 nm), augmented for the size of the primary and secondary FAB complexes (~15 nm on each side), and is nearly identical to that of Olivier[44] using improved STORM buffers and of Bates[45]. From this data we determine that the resolution in our experiments is at least ~22 nm when using primary and secondary FAB antibody labeling, and <22 nm when using primary antibodies directly labeled with a fluorophore.

**Averaging of STORM and EM signals**. Particle averaging was used to emphasize the symmetry in the distribution of DA EM and DAPs STORM signals. To average signals from nine appendages of the same centriole, the original STORM image was rotated for $40 \times n$ degrees ($n = 0, 1, 2, 3, 4, 5, 6, 7, 8$) around the physical center of the centriole (Supplementary Figure 10 and 11). The average, sum, and maximum projections were then generated from nine orientations. The same method was used to average signals from EM micrograms (Supplementary Figure 12).

**Electron microscopy**. For chemical fixation, cells grown on the coverslips or Transwell-Clear filters were fixed in 2.5% glutaraldehyde and 0.5% paraformaldehyde in 0.1 M Cacodylate buffer (pH7.4) and pre-stained with 1% osmium tetroxide and 1% uranyl acetate, dehydrated in graded ethanol series, and embedded in EMbed-812 resin. For High-pressure freezing, cells were grown on 6 mm sapphire discs, and frozen using the Leica EM ICE High Pressure Freezer, with 20% bovine serum albumin as cryoprotectant. For quick freeze substitution (QFS[46],), the sapphire discs were transferred to liquid nitrogen pre-cooled cryotubes containing QFS solution (0.1 g osmium tetroxide crystal in 4.5 ml 100% acetone, 0.25 ml 2% uranyl acetate in 100% methanol, and 0.25 ml ddH₂O), tubes were closed and placed into the metal box, and rocked at a speed of 60 cycles/min until the box reached room temperature. The sapphire discs were washed with 100% acetone for 3 × 10 min, infiltrated with a gradient acetone/Embed-812 resin mixtures and embedded in Embed-812 resin. Eighty or 120 nm-thick serial sections were sectioned, placed on the formvar-coated copper grids, and further contrasted with uranyl acetate and lead citrate. Imaging was performed using a Hitachi and FEI Spirit transmission electron microscope operating at 80 kV.

**Correlative STORM and EM**. To correlate 3D STORM signals with electron density signals, cells were immunolabeled. The target cell was first recorded in a multichannel wide-field mode as follows: 200 nm-thick Z-sections through entire cell volume were recorded to record XYZ coordinates of the centrioles within the target and surrounding cells. Only cells with all four C1-GFP signals in the same focal plane and with the older mother centriole perpendicular or nearly-perpendicular to the coverslip were considered for the analysis. The centrosome(s) in the target cells were then analyzed by STORM, and the position of the cell was marked on the coverslip by a diamond scribe[23]. Low magnification bright-field images of the target and the surrounding cells were then recorded to assist identification of the target cell during trimming and sectioning. Samples were post fixed in 2.5% glutaraldehyde and pre-stained with 1% osmium tetroxide and 1% uranyl acetate, dehydrated in a series of ethanol solutions, and embedded in EMbed-812 resin. Glass coverslips were dissolved, the target cell identified, and the region of interest was trimmed. Eighty nanometers-thick serial sections were sectioned, while preserving mother centriole's original orientation during serial sectioning. Sections were further contrasted and imaged as described in Electron microscopy paragraph.

**Alignment of STORM and EM images**. Images of the serial sections containing the target cell were taken at various magnifications and aligned using Photoshop (Adobe). The alignment of serial sections was performed in Photoshop, using centriole microtubules and other cellular landmarks as guides for the alignment[23] (Supplementary Figure 13). The extent of the sample shrinkage, which inevitably occurs during EM sample preparation[32] and imaging, was determined for each cell based on the difference in the distance between two objects laying in the same focal plane before and after EM analysis (Supplementary Figure 7b). The ratio between two measurements indicated the extent of the sample shrinkage.

To align wide-field and STORM images to the pre-aligned serial EM sections, we relied on a strict orthogonal orientation of procentrioles and mother centrioles and on Centrin1-GFP signals, which are localized in a small centriole lumen (smaller than the diffraction limit). This inherent symmetry and reproducibility of centrioles allowed us to accurately determine the centers of the centrioles. A vector through centers of the mother centriole and a procentriole was then drawn in both EM and wide-field recordings. The difference in the angle between two vectors represents a rotational angle for the alignment of the light and EM data. Please note that the rotational angle for wide-field and STORM recordings is the same, since the same centriole was imaged correlatively with the same lens, using the same magnification, and without moving the sample between two imaging modalities (wide-field and STORM). The light microscopy images were then fitted to the aligned serial EM sections (Supplementary Figure 13). The alignment was possible if three criteria were met: the mother centriole was associated with a procentriole, the mother centriole was vertically oriented to the coverslip and there was no introduced tilt in the sectioning plane during serial sectioning. The alignment was repeated several times for each sample. The alignment error (a standard deviation between three measurements) was within 5°.

**Superposition of STORM images from CLEM experiments**. To better visualize lateral arrangement of DAPs obtained in CLEM experiments, we superimposed CLEM STORM data as follows: First, we averaged the EM signals from individual centrioles used in CLEM experimets (Supplementary Method Fig. 12). Four averaged EM images were then lateraly aligned so that they were first centered and then rotated to obtain the maximal overlap of nine DA EM densities (Supplementary Figure 14). The average and minimum intensity projections of aligned EM images (Supplementary Figure 15a) revealed nine overlapping appendages. Next, the same rotation angles used to align EM images were used to rotate corresponding STORM images (Supplementary Figure 15b). Finally, the aligned STORM images were color coded (Fig. 3a), merged in one stack and using the maximum intensity projection function in Fiji, superimposed in various combinations (Fig. 3b).

**Electron tomography**. The tomographic data collection was carried out through SerialEM software[47] with a Philips Tecnai F20 electron microscope operating at 200 keV. Prior to data collection, the serial plastic sections were examined to assess specimen quality and to locate basal bodies with appendages using a 100 kV JEM 1400 electron microscope (JEOL). Then, 10 nm colloid gold beads (Sigma) were applied on both sides of sections as fiducial markers for tomographic reconstruction. The tilt series of electron micrographs were recorded on a 4k × 4k CMOS (Complementary Metal-Oxide-Semiconductor) based TemCam-F416 camera (TVIPS) with tilting angles ranged from −65° to +65° of 1° increment. Image alignment and structural map reconstruction were conducted using IMOD software[48] and Tomo3D software[49]. Briefly, the images of a tilt series were preprocessed to remove x-ray spots and then aligned roughly by cross correlation using IMOD. Fiducial markers were manually selected and refined for alignment. Tomograms (a-axis and b-axis tilt data sets) were separately reconstructed using Tomo3D with SIRT algorithm. Reconstructed tomograms were then combined using IMOD software. Segmentation of the tomogram and visualization of the distal appendage were carried out using the 3D interactive tool in Amira software (Thermo Fisher). Before segmentation, the "slicer" function in IMOD software was used to rotate the tomogram and look for a suitable orientation for convenient visualization and model building. Based on the orientation parameters in "slicer", the tomogram was rotated using programs in Bsoft package[50,51]. The re-orientated tomogram was imported into Amira software for structural visualization, model building, and movies making.

**Construction of RFP-HA Cep164 fragments and transfection.** The primers used for the construction of truncated versions of Cep164 were as follows: eRFP-F: acgtGCTAGCATGGTGTGTCTAAGGGCGAAGAG; eRFP-R: acgtAAGCTTTCCTCC ATTAAGTTTGTGCCCCAGTTTG; Full length-F: acgAAGCTTTACCCATACG ATGTTCCAGATTACGCTATGGCTGGACGACCCCTCC; Full length-R: acgGA ATTCTCAGAAGCGATACACCTTCACTCTG; ΔN99-F: acgAAGCTTTACCCAT ACGATGTTCCAGATTACGCTGCAAAGCTGTCAACTTCTGGG; ΔN297-F: acgAAGCTTTACCCATACGATGTTCCAGATTACGCTAAAGGGCGACAGGG AAGTGG; ΔN1200-F: acgAAGCTTTACCCATACGATGTTCCAGATTACGCT TGGGAAGAGGCCTCAGATGAG. The eRFP coding sequence without the stop codon was amplified by PCR using primers eRFP-F and eRFP-R and ligated into the pcDNA3.1(+) vector using *Nhe*I and *Hind* III restriction sites. Full-length Cep164 and its truncated fragments (ΔN99, ΔN297 and ΔN1200) were amplified from pEGFP-Cep164 (Nigg CW324), (Addgene plasmid # 41149[6]). HA sequence was inserted on N terminus during fragment amplification. Fragments were cloned into pcDNA3.1-eRFP using *Hind* III and *EcoR* I and expressed in cells using GenJet™ DNA transfection Reagent (Life Science's Service Center, Cat. #: M0014) alongside with 0.2 μM Cep164 siRNA oligonucleotide (CAGGTGACATTTAC TATTTCA (Dharmacon) following manufacturer's instructions. 2 days after transfection, cells were fixed and analyzed.

**Statistics.** Statistical differences between two samples was determine using a two-tailed Student's t-test in Excel for two unpaired samples. *P* values < 0.001 (marked as *** in image panels) were considered statistically different. Sample sizes are indicated in figure legends. A median line and upper and lower quartile is presented in box-and-whisker plots and dot-plots.

**Reporting summary.** Further information on experimental design is available in the Nature Research Reporting Summary linked to this article.

## Data availability

All data supporting the findings of this study are available from the corresponding author upon reasonable request. The source data underlying Figures 5a, 6a, and Supplementary Figure 5d are provided as a Source Data file.

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

## Acknowledgements

We thank members of Electron Microscopy Core at ATRF in Frederick for the assistance with High Pressure Freezing and substitution, and Drs. Catherine Sullenberger and Tomer-Avidor Reiss for critical reading of the manuscript. Electron tomography was enabled by the use of the Wadsworth Center's 3D-EM and EM Core Facilities. JL research was supported by the Intramural Research Program of the National Institutes of Health (NIH), National Cancer Institute, Center for Cancer Research, HS by the NIH (GM101026), AH by the NIH (R01GM114119), and MRM by the National Institutes of Diabetes and Digestive and Kidney Diseases (R01-DK108005).

## Author contributions

M.B. conducted most light microscopy experiments and analyzed the data. D.K. prepared samples for EM and tomography. S.S. and H.S. performed electron tomography, segmentation and model building. R.N. and M.R.M. contributed with mTEC cultures. L.E. and A.H. generated ANKRD26[KO] cell line. V.F. conducted SIM imaging. J.L. supervised the project, conducted EM and light microscopy imaging and analysis. All authors discussed the results. J.L., M.B. and D.K. wrote the manuscript.

## Additional information

**Competing interests:** The authors declare no competing interests.

