## [Peer Review File · Nature Communications]

Reviewers' comments:

Reviewer #1 (Remarks to the Author):

Bowler et al present a descriptive analysis of distal appendage proteins in mammalian cilia. The correlative STORM/EM is a nice approach but the morphological descriptions are not linked to any clear functional information. Moreover, fairly extensive STORM imaging of distal appendage proteins has been published recently by Yang et al. (2018, Nature Communications 9:2023). That study mapped many (but not all) of the proteins analysed in the present work. Newer findings are the tomography, CLEM, and cell cycle analysis, in which the distal appendage is described in more detail than in previous studies.

Aside from the tomogram, which is helpful, the videos are not very effective and contribute little useful information.

Overall, the work provides an improved description of the distal appendage, but does not reveal any significant new functional insights.

Reviewer #2 (Remarks to the Author):

The manuscript studied the distal appendage proteins at the end of centrioles by correlative super-resolution and electron microscopy. The overall quality of the 3D STORM images is good. However, there is too much important information missing, which makes me hesitate to support the publication at the current stage.

Main points:

(1) The manuscript used 3D STORM microscopy while all the images are shown in 2D projection mode, which means the axial information does not play a role. In my opinion, almost all the STORM imaging can be done in 2D and the resolution may be improved slightly without astigmatism.

(2) The authors reported the inner and outer diameter of each protein. However, there is no information about how to determine the diameters.

(3) The STORM image of Cep164 (Fig. 2b) is quite different from that of reference 27. What is the reason? The surface rendering of Cep164 is not convincing to me (Fig. 2c). I am not sure if it helps to be included.

(4) The authors used two-protein STORM to study the relations between each protein (Fig. 3b-d). I think two-color STORM is a better approach as we can differentiate the two proteins and we do not need to rely on correlative images to build the model. The alignment of the EM head densities does not look very good to me (Fig. 3b). Therefore, I am skeptical about the reliability of the superimposed images.

(5) The nine-fold symmetry is obvious in some images, but not as clear in others. I am wondering if the authors could do some particle averaging to further improve the quality so that the nine-fold symmetry may become more apparent.

(6) What is the precision of correlative imaging alignment? It is mentioned in the methods that the alignment was done manually, and only linear shrinkage was included. Considering the high resolution in both channels, I think it requires some better algorithms and a certain approach to quantify the precision.

Other points:

(1) Please quantify the resolution of the STORM imaging.

(2) Please specify the exact z positions in all the color maps. Labeling “top” and “bottom” is meaningless.

(3) Please report the laser intensity used for STORM imaging.

(4) Please add scale bars to the wide-field images in Fig. 2b.

Reviewer #3 (Remarks to the Author):

Loncarek – distal appendages

This is an outstanding paper that should be published in Nature Cell Biology without requiring any modifications. It meets a need in the centriole field to have an accurate description of the dynamic spatial relationships between the distal appendage proteins and uses super-resolution microscopy and EM-tomography to this end. The resulting description of the distal appendages is likely to become a standard reference work. The study has been extremely carefully executed and the findings very clearly presented. The paper will undoubtedly become very highly cited.

Comments from reviewers and point-to-point answers.

On behalf of all the co-authors, we thank the Reviewers for their comments, which helped us to improve the manuscript.

Reviewer #1 (Remarks to the Author):

Bowler et al present a descriptive analysis of distal appendage proteins in mammalian cilia. The correlative STORM/EM is a nice approach, but the morphological descriptions are not linked to any clear functional information.

Moreover, fairly extensive STORM imaging of distal appendage proteins has been published recently by Yang et al. (2018, Nature Communications 9:2023). That study mapped many (but not all) of the proteins analysed in the present work.

Newer findings are the tomography, CLEM, and cell cycle analysis, in which the distal appendage is described in more detail than in previous studies.

Aside from the tomogram, which is helpful, the videos are not very effective and contribute little useful information.

Overall, the work provides an improved description of the distal appendage but does not reveal any significant new functional insights.

Answer:

1. We have followed the suggestion of the reviewer and removed supplemental videos.
2. In this version of manuscript, we are highlighting the links between structure and function in the last chapter. However, we would like to emphasize that this manuscript was envisioned as a detailed analysis of structural and dynamic properties of appendages, with the idea that it will provide so needed and reliable structural framework for ongoing and future functional studies. It is also critical to understand that Yang study builds the appendage model exclusively based on two color STORM data, which cannot provide the insight into the localization of appendage proteins in relation to centriole microtubules and appendage EM densities. Thus, the model proposed in Yang's study remains hypothetical.

Reviewer #2 (Remarks to the Author):

The manuscript studied the distal appendage proteins at the end of centrioles by correlative super-resolution and electron microscopy. The overall quality of the 3D STORM images is good. However, there is too much important information missing, which makes me hesitate to support the publication at the current stage.

We would like to thank this Reviewer on critical comments, which help us to improve the manuscript. Especially for the excellent suggestion to average STORM signals. We apologize for omitting some relevant methodological descriptions in the first version of the manuscript. We corrected that and in this version of the manuscript we provide 9 Supplementary Methods Figures. We also introduced two additional Supplementary Figures (current Supplementary Fig. 2 and 4), to illustrate the symmetry and reproducibility of STORM signals, and to further reinforce the conclusions of CLEM analysis by two-protein STORM. We hope that you will find all the points you have raised appropriately answered.

Main points:

(1) The manuscript used 3D STORM microscopy while all the images are shown in 2D projection mode, which means the axial information does not play a role. In my opinion, almost all the STORM imaging can be done in 2D and the resolution may be improved slightly without astigmatism.

Answer: We agree that performing 2D STORM could slightly improve the resolution and could have been used for some analyses. 2D STORM would have also been technically less challenging to perform.

However, at the onset of this study we have made a careful evaluation of the initial data obtained by 2D and 3D STORM. 3D analysis provided far superior and more insightful information than the 2D analysis.

To appreciate our decision to use 3D analysis, it would be important to take into consideration the biology of the centrosome and the fact that the orientation of centrioles in the population of cells is random. Having the information about the spatial orientation of centrioles by using 3D STORM was critical for the following reasons:

One: Only by knowing how centrioles were oriented with respect to the imaging plane, allowed us to properly interpret morphological changes during initial appendage assembly on younger mother centrioles, and during remodeling of appendages in mitosis. This would not be possible by using 2D STORM, because a centriole tilt would be difficult and even impossible to gauge. It is relatively straightforward to interpret STORM images of mature centrioles during interphase, because they were expected to exhibit a toroid organization of specific diameters (predetermined by SIM). However, to accurately describe the initial stages of appendage formation and their dynamics during mitosis, when appendage signals are of a random number and at variable levels, 3D STORM analysis was critical.

Two: In CLEM experiments, STORM imaging in 3D allowed us to be assured that the centrioles subsequently analyzed by EM were indeed perpendicular to the coverslip. After 3D imaging, we inspected every centriole in a 3D viewer, to determine their possible tilt with respect to the coverslip. This would be impossible by using only 2D analysis. Obviously, even a small centriole tilt would complicate or even impede the subsequent STORM/EM alignment, so the 3D imaging allowed us to select only centrioles perpendicular to the coverslip for subsequent EM analysis. In our estimation only ~5% of imaged centrioles for CLEM were properly orientated and were further embedded for CLEM. Thus, without 3D analysis, we could not have been sure that we preserved the original orientation of the centrioles throughout EM analysis.

Three: Throughout the manuscript, we preserved the original Z-depth color coding scheme to present the STORM data. We learned that using color coding greatly facilitates interpretation of the data, especially by the colleagues outside of the field and those not so familiar with imaging.

From all the above reasons, we resorted to the 3D analysis and we believe that that was the correct decision.

(2) The authors reported the inner and outer diameter of each protein. However, there is no information about how to determine the diameters.

Answer: We are now illustrating how STORM signals were measured in Supplementary Material Figure 2. Briefly, to determine the outer and inner diameter of STORM, the signals were first fitted into two circles, and the diameter of the outer and the inner circle was measured.

(3) The STORM image of Cep164 (Fig. 2b) is quite different from that of reference 27. What is the reason?

Answer: Throughout the manuscript, Cep164 imaging has revealed that Cep164 occupies a wider surface than other DAPS, and that is organized in the loop-like pattern. Such pattern has been detected on the appendages of three cell lines. We do not know for certain why such loops have not been detected in ref. 27. We can only list several possible reasons for this discrepancy.

Different antibodies: In ref 27 (Yang et al., 2018) uses two antibodies; a rabbit antibody targeting Cep164 middle region (Novus Biologicals) and a goat antibody targeting the N-terminal region (Santa Cruz, sc-240226). In our study, we used Cep164 rabbit antibody from Proteintech (22227-1-AP) to detect N-terminal portion of Cep164. In comparison with sc-240226 antibody, which is not effective in immune precipitation or in western blots, Proteintech antibody performs in all assays. This just to illustrate the difference between two reagents. Although we successfully used sc-240226 antibody in the past for immunostaining in wide-field mode (Kong et al, JCB, 2015), it is impossible for us to validate this

antibody for STORM, since it has been discontinued a while ago and is unavailable.

However, we tested currently available sc- 15403 antibody, which targets N terminus Cep164. In our hands, this antibody allowed us to occasionally detect Cep164 loops, (Figure below) but even after numerous attempts of optimization, the signal appeared inconsistent. This antibody also performed poorly after direct labeling it was not further used for STORM.

Finally, based on our analysis of truncated versions of Cep164 protein, Cep164 is organized in a radial fashion with its N-termini localized in a wider loop than its middle region. Therefore, some differences in the appearance of the signal could be expected and the loop-like organization of Cep64 might not have been apparent if

middle portions of Cep164 were labeled. We cannot comment further on this matter, as it is not clear which antibody was used for which panel in ref 27.

Direct antibody labeling. Another difference is that in our study, we used the primary Cep164 antibody directly conjugated with CF647 fluorophore. Thus, we significantly reduced the size of the antibody-fluorophore complex during immune detection, which might have contributed to the labeling of additional set of Cep164 epitopes.

Fixation method. It is known that the fixation method can affect the epitope and its recognition by the antibody. In our work, we first pre-fixed cells in 1.5% formaldehyde 4 min, followed by a post fixation in cold 100% methanol at -20°C for additional 4 min. In ref 27, Yang et al use either 4% formaldehyde or cold Methanol for fixation (the duration of fixation is unspecified).

The surface rendering of Cep164 is not convincing to me (Fig. 2c). I am not sure if it helps to be included.

Answer: We followed the recommendation and removed the surface rendering data from Fig. 2 (previous Fig. 2b). We re-arranged other panels accordingly.

(4) The authors used two-protein STORM to study the relations between each protein (Fig. 3b-d). I think two-color STORM is a better approach as we can differentiate the two proteins and we do not need to rely on correlative images to build the model.

Answer: We will need to respectfully disagree with the Reviewer. The correlative analysis was necessary to build the model and to determine the exact localization of appendage proteins with respect to the centriole barrel and EM densities. At the same time, it revealed how various appendage proteins are localized with respect to each other. Superposition of the STORM images from CLEM as illustrated in Fig. 3b, only served to facilitate the visualization of their relative localization. Two-protein STORM was then used to ascertain that the relative positions between proteins determined by CLEM were correct, and to confirm that the alignment of the EM and STORM data was precise.

If the signals from two proteins can be unambiguously separated, the two-protein STORM is a superior method to determine their spatial relationship. It guarantees that no chromatic shift or any other shift due to the alignment, for instance, will be introduced during imaging. Thus, it is superior in its precision over two-color STORM. Another strong reason why we did not resort to two-color STORM in this analysis is that two color STORM works well if both proteins are abundant. However, appendage proteins are present at centrosomes at quite low stoichiometry (Bauer et al, EMBO, 2016). Our experience is that two-color STORM compromised the detectability of at least one protein, but often both (published STORM buffers did not work optimally for both fluorophores). We have discussed this matter in length with the colleagues from several microscopy cores, whom had similar experience with two-color STORM.

The alignment of the EM head densities does not look very good to me (Fig. 3b). Therefore, I am skeptical about the reliability of the superimposed images.

Answer: We now provide Supplemental Material Fig. 7, 8 and 9, where we provided a detail description of the alignment and superposition of EM and STORM signals. We hope that this will convince the Reviewer that the superposition of the data as shown in Fig. 3B is reliable. As we emphasized in our answer to the Specific point 4, the superimposition of the data serves only facilitate the interpretation. No conclusions have been derived from it, as the answer about the relative positions of DAPS is self-evident from the CLEM and two-protein STORM experiments (which are two independent methods which yielded identical conclusions).

(5) The nine-fold symmetry is obvious in some images, but not as clear in others. I am wondering if the authors could do some particle averaging to further improve the quality so that the nine-fold symmetry may become more apparent.

Answer: We would like to thank the Reviewer for this excellent suggestion. We averaged STORM signals belonging to the same centriole, which further accentuated their nine-fold symmetry. Averaged signals are now included alongside the original data in Fig. 2b and c, 6g. Also, we have replaced Cep164 STORM data from Fig. 2b with another one, in which, we hope, the Cep164 loops will be more apparent. We have also added a new Supplemental figure (Current Supplementary Fig. 2), where we show more examples of averaged CCDC41, FBF1, SCLT1, and Cep164 signals, to illustrate their nine-fold symmetry and uniformity. More examples are available if needed. The method for signal averaging is now described in materials and methods and is illustrated in Supplementary Material Fig. 4 and 5. Similar method of averaging centriole EM signals have been used previously by Paintrand et al., J Struct Biol., 1992.

Please note that the averaging cannot be applied to all STORM signals. The appendages in the state of the assembly and disassembly (young centrioles and around mitosis), naturally show greater qualitative and quantitative differences and often lack nine-fold symmetry (Fig. 5, younger centrioles, 6d- prophase, 6c-prometaphase, 6h-early G1).

Two outer proteins, Cep164 and ANKRD26, and proteins associated with them such as TTBK2 (already published outer appendage protein with no obvious nine-fold pattern) are expected to be in flux, as we hypothesize that they mediate various cell cycle signals, respond to centrosome environment... Thus, it is not surprising that Cep164 signals belonging to one centrosome are somewhat variable in shape and that the loops are not detectable around all appendages. This is exactly what the EM analysis would predict. EM data shows that the appendages of one centriole exhibit certain variability in shape, intensity of their EM heads and variability in the density of the surrounding material. Distances between appendage heads can also slightly vary within the centrosome (please see Fig. 1, Fig. 6, 1e tomogram movie, quantification of EM signals in Fig. 6f). The fact that we were able to detect this level of symmetry for outer proteins such as Cep164, and FBF1 is, in our view, quite remarkable.

(6) What is the precision of correlative imaging alignment? It is mentioned in the methods that the alignment was done manually, and only linear shrinkage was included. Considering the high resolution in both channels, I think it requires some better algorithms and a certain approach to quantify the precision.

Answer: We agree with the Reviewer that it would be great to have precise algorithms for aligning STORM and EM images. Unfortunately, we are not aware of any algorithm that would be suitable for this study. So, we relied on our longstanding and extensive experience in correlative light/electron analysis of the centriole (Loncarek et al., NCB, 2007, Kong et al., Mol Cell Biol., 2015, Shukla et al., Nat. Commun, 2015., Khire et al, Curr. Biol., 2016, Wang et al., Elife, 2017., Fishman et al., Nat. Comm. 2018., Gouveia et al., J cell Sci., 2018).

We have added more detailed description of our alignment procedure in Material and Methods chapter and introduced Supplementary Methods Fig. 4, which illustrates the accuracy of our alignment.

The alignment of serial sections was performed in Photoshop, using centriole microtubules and other cellular landmarks as guides for the alignment. This method for centriole alignment has been widely used for centriole alignment and is a routine procedure in our laboratory. The shrinkage coefficient that occurs during sample preparation was calculated based on the difference between two objects laying in the same focal plane before and after EM analysis, as described in Materials and Methods and in current Supplementary Method Figure 3.

To align wide-field and STORM images to the pre-aligned serial EM sections, we used a unique feature of duplicated centrioles. After centriole duplication procentrioles of S and G2 phases of the cell cycle are positioned perpendicularly to the mother centriole. In addition, Centrin1-GFP is localized in a relatively small centriole lumen, which is smaller than the diffraction limit of conventional microscopy. This inherent symmetry and reproducibility of centrioles allowed us to accurately determine the centers of the centrioles (the centers of mass for Centrin-GFP signal). A vector through centers of the mother centriole and a procentriole was then drawn in both EM and wide field recordings. The difference in the angle between two vectors (measured either in Photoshop or Fiji) represents a rotational angle for the alignment of the light and EM data. Please note that the rotational angle for wide-field and STORM

recordings is the same, since the same centriole was imaged correlatively with the same lens, using the same magnification, and without moving the sample between two imaging modalities (wide-field and STORM). The light microscopy images were then fitted to the aligned serial EM sections.

This alignment procedure is possible only if three criteria are met: A mother centriole needs to be duplicated, a mother centriole needs to be vertically oriented to the coverslip (hence the need for 3D STORM) and no tilt in the sectioning plane can be introduced during serial sectioning. If these criteria are met, the alignment of EM and STORM images is straightforward, reproducible and independent on the initial rotation of the serial sections. In addition, in some cases, we used the coordinates of Centrin1-GFP signals belonging to another duplicated centriole lying in the same focal plane, to independently calculate the rotational angle for the light microscopy images.

The alignment was repeated several times for each sample and we calculated that the alignment error was $<5^\circ$ degrees (2.5° from the center of the appendage EM density). Several samples per protein were analyzed. Please note, that centriole is a nine-fold symmetrical structure with the rotational symmetry of 40° . The model would not significantly change even if the error of alignment were much bigger.

The accuracy of our alignment was further conformed by two-protein STORM analysis, which revealed the same mutual organization of DAPS as CLEM analysis.

Other points:

(1) Please quantify the resolution of the STORM imaging.

Answer: The resolution of the STORM imaging has been quantified and this information is now included in Supplemental Material and Methods.

(2) Please specify the exact z positions in all the color maps. Labeling “top” and “bottom” is meaningless.

Answer: We have better defined the color coding scheme.

(3) Please report the laser intensity used for STORM imaging.

Answer: The output laser intensities are now reported in Material and methods. The power of imaging laser was 150 mW out of the fiber and ~ 90 mW before objective lens.

(4) Please add scale bars to the wide-field images in Fig. 2b.

Answer: Scale bar has been added.

Reviewer #3 (Remarks to the Author):

Loncarek – distal appendages

This is an outstanding paper that should be published in Nature Cell Biology without requiring any modifications. It meets a need in the centriole field to have an accurate description of the dynamic spatial relationships between the distal appendage proteins and uses super-resolution microscopy and EM-tomography to this end. The resulting description of the distal appendages is likely to become a standard reference work. The study has been extremely carefully executed and the findings very clearly presented. The paper will undoubtedly become very highly cited.

Answer: We are delighted and would like to thank this Reviewer for acknowledging our work.

Reviewer #1 (Remarks to the Author):

The revisions have improved the manuscript, but it remains a detailed description of distal appendages using a nice application of correlative STORM/EM, which will be most accessible to centriole specialists. The authors acknowledge that their goal was for this work to serve as a framework for future functional studies.

Symmetry averaging could be a useful addition, but 9-fold symmetry was applied to individual images rather than aligning and averaging multiple images as is typical. The step by step description of rotational superposition is reminiscent of descriptions of Markham rotation from the 1970s- image alignment and rotational averaging are now standard methods.

Reviewer #2 (Remarks to the Author):

The manuscript has been improved significantly after revision. The averaged STORM images clearly reveal the nine-fold symmetry of the proteins. The missing detail information is included in Supplementary Material. The authors have addressed all my concerns and I am happy to support the publication.